# Bats as ecosystem engineers in iron ore caves in the Carajás National Forest, Brazilian Amazonia

**Luis B. Piló**[1†]**, Allan Calux**[2]**, Rafael Scherer**[3]**, Enrico Bernard**[1]*

**1** Departamento de Zoologia, Laboratório de Ciência Aplicada à Conservação da Biodiversidade, Universidade Federal de Pernambuco, Recife, PE, Brazil, **2** Carstografica–Karst Applied Research Centre, Campinas, SP, Brazil, **3** Grupo Espeleológico de Marabá, Marabá, PA, Brazil

† Deceased.
* enrico.bernard@ufpe.br

**Data Availability Statement:** All relevant data are available on Zenodo (DOI: 10.5281/zenodo.6875116).

**Funding:** LB Piló received funds from speleological compensation TCCE – ICMBio/Vale 01/2018. The funders had no role in study design, data collection

## Abstract

Ecosystem engineers are organisms able to modify their environment by changing the distribution of materials and energy, with effects on biotic and abiotic ecosystem components. Several ecosystem engineers are known, but for most of them the mechanisms behind their influence are poorly known. We detail the role of bats as ecosystem engineers in iron ore caves in the Carajás National Forest, Brazilian Amazonia, an area with > 1,500 caves, some holding ~150,000 bats. We analyzed the chemical composition of guano deposits in bat caves, radiocarbon-dated those deposits, and elucidated the chemical mechanisms involved and the role the bat guano has on modifying those caves. The insect-composed guano was rich in organic matter, with high concentrations of carbon, nitrogen, phosphorus pentoxide and ferric oxide, plus potassium oxide, calcium and sulfur trioxide. Radiocarbon dating indicated guano deposits between 22,000 and 1,800 years old. The guano pH was mainly acid (from 2.1 to 5.6). Percolating waters in those bat caves were also acid (pH reaching 1.5), with the presence of phosphate, iron, calcium, nitrate and sulfate. Acid solutions due to guano decomposition and possible microbial activity produced various forms of corrosion on the caves´ floor and walls, resulting in their enlargement. Bat caves or caves with evidence of inactive bat colonies had, on average, lengths six times larger, areas five times larger, and volumes five times bigger than the regional average, plus more abundant, diversified and bigger speleothems when compared with other caves. In an example of bioengineering, the long-term presence of bats (> 22,000 years) and the guano deposits they produce, mediated by biological and chemical interactions over millennia, resulted in very unique ecological, evolutionary and geomorphological processes, whose working are just beginning to be better understood by science. However, the current expansion of mineral extraction activities coupled with the loosening of licensing and cave protection rules is a real conservation threat to the bat caves in Carajás. The destruction of those caves would represent an unacceptable loss of both speleological and biological heritage and we urge that, whenever they occur, bat caves and their colonies must be fully protected and left off-limits of mineral extraction.

and analysis, decision to publish, or preparation of the manuscript.

**Competing interests:** The authors have declared that no competing interests exist.

## Introduction

Ecosystem engineers are organisms able to modify their environment usually by changing the distribution of materials and energy, with effects on both the biotic and abiotic components of that ecosystem [1,2]. Ecosystem engineers may create, maintain or modify habitats by changing the chemical and/or physical composition of substrates and structures in those habitats. Engineering species can be basically divided into autogenic engineers, which change the environment by modifying themselves, or allogenic engineers, whose influence on the environment happens by physically or chemically transforming the state of living or non-living materials, primarily by mechanical means [1].

Since the definition of the concept of ecosystem engineers by Jones et al. [1], an increasing number of examples have been presented, from earthworms, termites, lepidopterans, corals and bryozoans, to beavers, porcupines and humpback whales [e.g. 3,4]. Earthworms, for example, are able to decompose organic matter in the soil, interfering in the dynamics of soil carbon in forest and savannas ecosystems [3]. The mutualism between corals and their intracellular photosynthetic zooxanthellae allows the capture and use of calcium carbonate for the growth of their skeleton structure, creating an environment essential for hundreds of other marine species [5]. The construction of tunnels and nests by termites moves enormous amounts of soil and enhances bioturbation processes essential for soil formation [6]. Beavers and giant armadillos are examples of land mammals whose damming or burrowing habits, respectively, can structurally modify the surrounding landscape, affecting several other species [7,8].

However, although several examples of ecosystem engineers are known, for most of them the mechanisms behind their influence on the ecosystem *per se* are just superficially described [4,5]. A better understanding on how organisms modify the physical and chemical abiotic factors may help ecologists to better predict effects of ecosystem engineering on biogeochemical processes, on species distributions, or on variations of ecosystem engineers across environmental gradients [5,9–12]. But, maybe more importantly, ecosystem engineers and their associated engineering are examples of ecological-evolutionary interactions with potential consequences on different levels of biological organization, from population biology to landscape and community ecology, and between physiology and ecosystems [5,7,9]. Moreover, the interactions involving ecosystem engineers are frequently unique, fragile and not easily reproducible or compensable. Therefore, the identification of ecosystem engineers may also have direct implications on management measures related to human impacts, especially when endangered species, important ecosystem processes or environmental services essential to humans are involved [5,6,13].

Caves housing exceptionally large bat populations–frequently surpassing hundreds of thousands of bats–are known as bat caves [14–17]. There is no such a universally accepted definition on how to classify a bat cave, neither there is a threshold number of bats above which a cave could be labelled as a bat cave [see 17]. In some bat caves, the body heat of bats and the guano decomposition generates a very high temperature, often $> 34°C$, making them also hot caves [15]. Such hot caves are known in the Antilles [18], Cuba [19–21], Puerto Rico [22], Mexico [23], Venezuela [18,24], and Brazil [25–27]. Bat caves are ecologically unique environments, where biological, chemical and physical interactions occur in ways that are still superficially known to science. But a distinguished characteristic of bat caves is the presence of extensive volumes of guano [17,28,29], recognized as a key element for such interactions [17].

Here we provide evidence of the role of bats as ecosystem engineers in iron ore bat caves in the Carajás National Forest (hereafter CNF), in the Brazilian Amazonia, an area with $> 1.500$ caves, some housing $> 150,000$ bats. We hypothesized that cave structures and corrosion processes observed in those bat caves were unique and intrinsically mediated by the presence of

such large bat populations and their associated guano and, therefore, would not be observed in caves with no evidence of the presence of bats. Thus, we a) analyzed guano deposits in bat caves to identify their chemical composition, b) radiocarbon dated those deposits to estimate for how long bat colonies have been present, and c) elucidated the chemical mechanisms involved and the role the bat guano may have had on the structuring of those habitats, particularly in the significant chemical deposition of speleothems and in the corrosion processes on the floors and walls of the caves. In this process, we analyzed active bat caves (i.e., caves currently housing large bat colonies and guano), inactive bat caves (i.e., caves without current large colonies, but with evidence of past presence of bats and guano) and compared the results from those caves with a large dataset taken from other regional caves.

## Material and methods

### Study area

The CNF (6˚04'44"S, 50˚10'38"W) located in the southeast of Pará State, Brazil (Fig 1), is one of the largest mineral deposits in the world [30–32]. The climate in the Serra dos Carajás region fits Köppen's classification as humid tropical Aw, with a dry season from June to

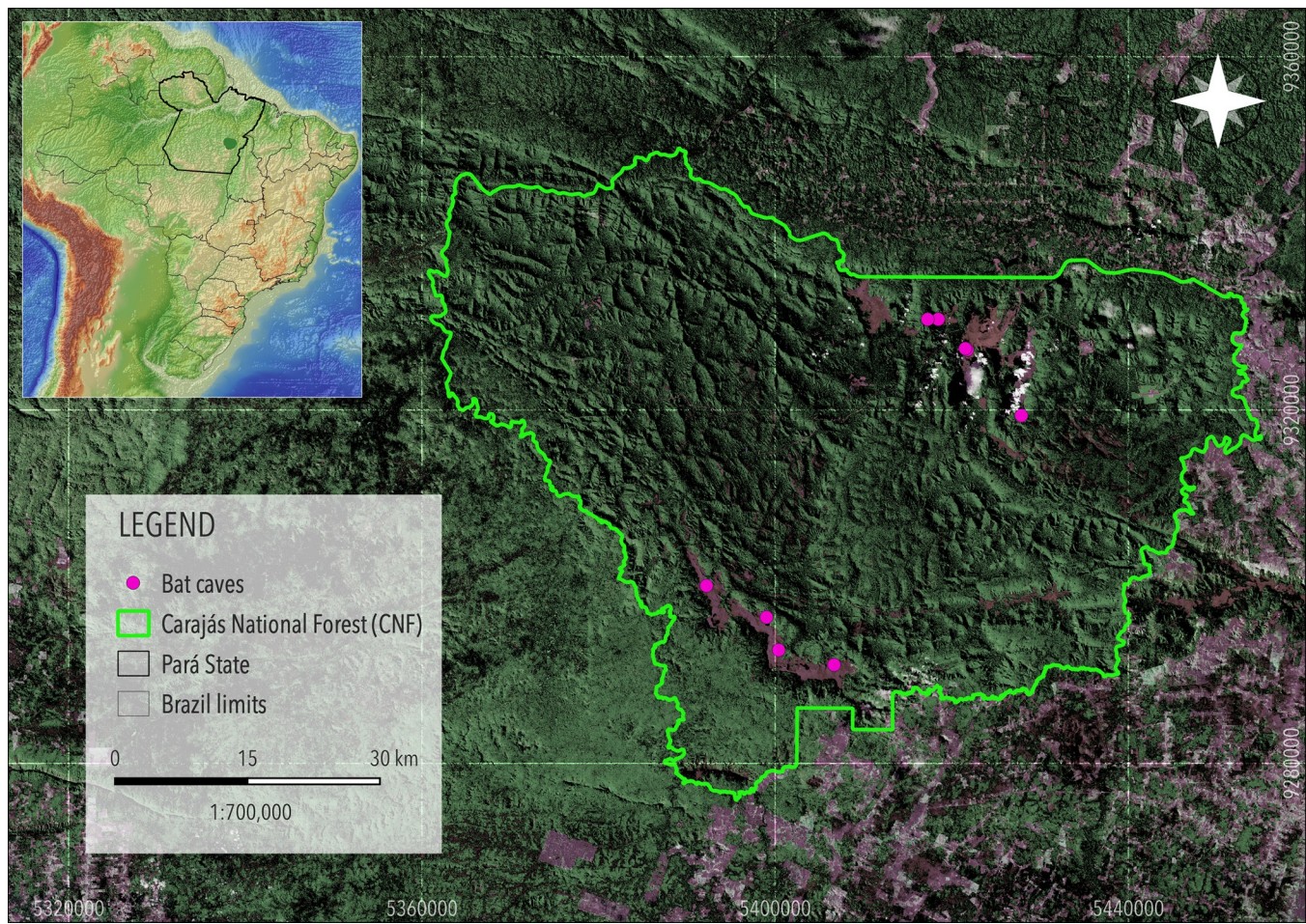

**Fig 1. The Carajás National Forest, in southeastern Pará State, Brazilian Amazonia.** Darker areas are *terra firme* forest, while lighter areas represent deforested areas. The bat caves sampled are grouped in two major iron ore extraction areas, Serra Norte (top) and Serra Sul (bottom). Satellite image: NASA Landsat Program, 2021, Landsat 7 ETM+, scene LE72240642021357ASN00, LE07_L1GT_224064_20211223_20220118_01_T2, SLC-Off, USGS, 12/23/2021.

September (average rainfall of < 60 mm in the driest months) and a rainy season from November to April, with ca. 1,500–2,000 mm of rain annually [33,34]. The average annual temperature is 23.8˚C (from 12.2˚C to 37˚C). The relative humidity remains around 70%, with 95% from October to May [33,34].

Geologically, the Carajás Formation stands out as a Neoarchean metavolcanosedimentary sequence within the Grão Pará Group. It is composed of banded iron formations represented by jaspilites. The iron formation is covered by iron-rich breccia generically known as *canga*, which act as caprock on plateau tops, regionally represented by the Carajás Ridge. On these *canga* covers, particularly at the top of the mountains, savannas stand out, surrounded by an exuberant ombrophilous forest. The xerophilous savanna vegetation represents a unique ecosystem in Amazonia, with high levels of endemism [35]. The forest parts, on the other hand, represents the largest remnants of tropical forest preserved in southeastern Amazonia.

The Carajás region is one of the best inventoried speleological provinces in Brazil and over 1,500 caves have already been identified in the Carajás National Forest. This is one of the largest concentration of caves in Brazil. The caves are located at the foot of scarps situated in different landscape settings, including at the edges of ponds, on scarps at the top of the plateaus, as well as in the colluvial foot slopes of ridges. The caves have developed in the inner part of the ferruginous breccias, within the banded iron formations, and at contact points between them. Dripping is significant inside the caves due to their proximity to the surface and porosity of the *canga* [36]. Some caves function as springs. Temporary drainage channels can also occur. The clastic deposits found in the caves are predominantly autogenic consisting of hematite clasts originating from the banded iron formations, *canga*, and sometimes altered mafic rocks [37].

## Bats and bat caves in Carajás

At least 8 families, 46 genus and 76 bat species have been already recorded in Carajás, with 23 of those species recorded in caves [38]. Currently, among the ~1,500 caves known in the Carajás National Forest, only five of them currently have large bat colonies, characterizing them as bat caves. Some of those caves hold populations estimated at > 150,000 bats each (E. Bernard, pers. observ.). The large colonies in Carajás´ bat caves are formed mainly by *Pteronotus gymnonotus* and *P. personatus*, two strictly insectivorous bats of the Mormoopidae family, and the guano in these caves is mainly composed of insect remains. No other sources of guano (e.g. from swiftlets or oil birds) are present in the caves we studied, so the guano volumes observed are unequivocally produced mainly by *Pteronotus* bats.

*Pteronotus gymnonotus* is a medium-sized bat (8 to 16 g), whose distribution ranges from southeastern Mexico through Central and South America south to northeastern Bolivia and central Brazil [39]. *Pteronotus personatus* is smaller (5 to 9 g), ranging from Mexico to central and northeastern Brazil [40]. Studies conducted in bat caves in Northeastern Brazil have shown that insect consumption may vary from 0.6 to 2.5 g/bat for *P. gymnonotus* (~5 to 20% of their body weight) and 0.8 to 2.0 g/bat for *P. personatus* (~10 to 28% of their body weight) and, in certain situations, guano deposition on the cave floor may reach up to 738 g/m$^2$/96h [29]. Therefore, when present, those bats play a fundamental role on the guano input in any bat cave.

## Cave and speleothem characterization

We selected 41 caves for an in deep analysis in Carajás, including three active bat caves (i.e., caves currently harboring large bat colonies estimated at tens of thousands of bats), seven inactive bat caves (i.e., caves with evident signs of past presence of bats, such as ancient guano

deposits–S1 Table), and 31 caves where other samples were collected for comparison (S2 Table). Sampling and collections were granted by permit SISBIO 64633–1, 64633–2 and 64633–3. Some of the active bat caves had guano deposits up to ~90 cm deep, which stood out from the bare rock. Whenever possible, bat caves had their length, area and volume measured or estimated (some from previous cave surveys), and those were then compared with measurements from a dataset of 1,309 caves in the same area. This database was compiled by the authors from different sources, including licensing reports and consultancies. Speleothems in the bat caves were described based on shape, size, color, and location (ceiling, walls or cave floor). Speleothem and guano composition was also assessed: Between May and December 2019, 108 samples from 18 stalactites, stalagmites and crusts were collected for mineralogical (88 samples) and chemical (20 samples) analysis, performed at the University of São Paulo´s Escola Politécnica, in Brazil (see Supporting information). We used the powder method and a Panalytical X-ray diffractometer (X'Pert model with an X'Celerator detector) for mineralogical analyses, and an X-ray spectrometer manufactured by Malvern Panalytical (Zetium model) for the chemical analyses (oxides).

## Bat guano analyses

Twenty-two guano samples were collected directly from the cave floor and from excavated trenches and sent for analysis at the Laboratório de Fertilizantes, Corretivos e Resíduos Orgânicos laboratory of the Escola Superior de Agricultura Luiz de Queiroz—ESALQ/USP, a national reference for chemical analysis of soil in Brazil (see Supporting information). The methods used included: spectrophotometry and vanadomolybdenic solution for phosphorus ($P_2O_5$); flame photometry for potassium ($K_2O$) and sodium (Na); the gravimetric technique with precipitation of barium sulphate for sulphur (S); and the extraction with HCl for atomic absorption photospectrometry for calcium (Ca), magnesium (Mg), copper (Cu), manganese (Mn), zinc (Zn), iron (Fe).

In laboratory conditions, a 0.01 mol L-1 CaCl2 solution was used to determine the guano pH. Six samples from each cave were collected on the surface of the guano deposits in six different caves and 16 samples were collected inside the trenches of three caves: in cave N3-0023, samples were taken at 65 cm deep (sample P1) and 45 cm deep (P2); in cave N5SM2-0099 at 95 cm deep; and in cave S11C-0041 at 30 cm deep. After opening the trenches, homogeneous morphological layers were defined, including color (Munsell Soil Color), texture and structure. After the description of the layers, a sample of each layer was removed with a spatula and placed in plastic bags. The main constituents of each layer were analyzed in Carl Zeiss Stemi 2000C microscopes.

## Analysis of water pH and chemistry

We used the portable YSI EcoSense 100A with field probe (SKU 605377) for measurements of pH of shallow waters. We took 56 pH measurements of circulating waters in the caves and 10 measurements of surface waters (ponds and rivers). The chemistry of the circulating waters in the caves was analyzed using a portable photometer YSI Ecosense 9300. Fourteen samples were analyzed in two inactive bat caves (N3-0023 and N4WS-0072), and two caves without bats (N5S-0011 and N5S-0012). The following parameters were analyzed: ammonia $NH_4$, Fe, $PO_4$ and $NO_3$.

## Radiocarbon dating

Eight guano samples and two stalagmite samples were used for $^{14}C$ dating, using accelerator mass spectrometry (AMS) technique, carried out at the Beta Analytic Laboratory

([www.radiocarbon.com](www.radiocarbon.com)), in Miami (USA) (see Supporting information). The guano samples were collected between May and December 2019 in trenches opened in one active bat cave (N5SM2-0099) and four inactive bat caves (N3-0023, S11B-0094, S11C-0041, and N5S-0063). Samples were taken mainly at the base of the guano deposit, in order to obtain the age of onset of deposition, but samples were also taken at the top of the sedimentation and at intermediate positions. In cave N4WS-0067, a sample of guano was obtained inside a small cavity near the cave floor. In cave S11B-0094, one stalagmite consisting of phosphates was collected, which provided two samples for dating the organic matter trapped in, one at the top and one at the bottom of the speleothem. The choice for this collection was made for the analysis of possible layers of guano deposited during the process of formation of the stalagmite.

## Results

### Cave dimensions

Considering the regional sample of 1,309 caves (with the nine bat caves included), cave length varied from 3.2 up to 460.0 meters, area varied from 1.2 up to 1,836.0 $m^2$, and the volume from 3.0 up to 3,823 $m^3$. Measurements for the four active and six inactive bat caves alone were: length varied from 52.4 up to 365.6 meters, area from 351.2 up to 1116.3 $m^2$, and volume from 765.0 up to 2254.0 $m^3$. Although there was overlap among dimensions (S1 Fig), caves currently harboring bat colonies (active bat caves) and those with signs of past presence of such colonies (inactive bat caves) had, on average, length 6.1 times larger, area 5.3 times larger, and volume 5.8 times bigger than the regional average (Table 1). Considering the regional sample (n = 1,309 caves), the longest cave reached 460.0 meters but with no corrosion processes like those observed in active or inactive bat caves (Fig 2). Moreover, active and inactive bat caves had more abundant, diversified and bigger speleothems when compared with the regional dataset. Speleothems in caves with no expressive guano deposits were of local occurrence and small, while the greater diversity, abundance and size of speleothems was found in the bat caves, with stalactites, stalagmites, columns, coralloids and crusts sometimes almost entirely covering the floor, walls and ceilings (Fig 2C and 2E).

**Bat guano analyses.** The guano analyzed in the CNF caves was mainly composed of organic material with a granular structure and the remains of crushed insect exoskeletons (Fig 3). The guano had an acidic pH, which ranged from 2.2 to 5.4. Profile N3-0023-P2 was the most acidic (Table 2; Fig 3). In the N5SM2 profile, there was a decrease in pH with depth. In the other profiles, the pH variations with depth are small. The total organic matter ranged from 27.7% to 58.4%. These maximum and minimum values were recorded in surface sediments.

**Table 1. Comparison of length (in meters), area (in square meters) and volume (in cubic meters) between bat caves and non-bat caves in the Carajás National Forest, Brazilian Amazonia.**

|  | Length (m) | | Area (m²) | | Volume (m³) | |
|---|---|---|---|---|---|---|
|  | Bat caves | Regional sample | Bat caves | Regional sample | Bat caves | Regional sample |
| n | 10 | 1309 | 10 | 1309 | 10 | 1309 |
| average | 198.2 | 32.3 | 650.6 | 121.5 | 1320.2 | 224.4 |
| standard deviation | 72.0 | 41.2 | 261.3 | 187.6 | 522.2 | 408.7 |
| median | 189.8 | 18.0 | 647.8 | 54.0 | 1020.0 | 72.8 |
| 95th percentile | 306.6 | 108.0 | 1066.1 | 492.4 | 2076.0 | 977.7 |
| Q3 + 1.5IQR | 336.6 | 68.4 | 1482.3 | 274.5 | 3032.2 | 498.0 |
| Average + 2sd | 342.2 | 114.6 | 1173.1 | 496.7 | 2364.7 | 1041.7 |

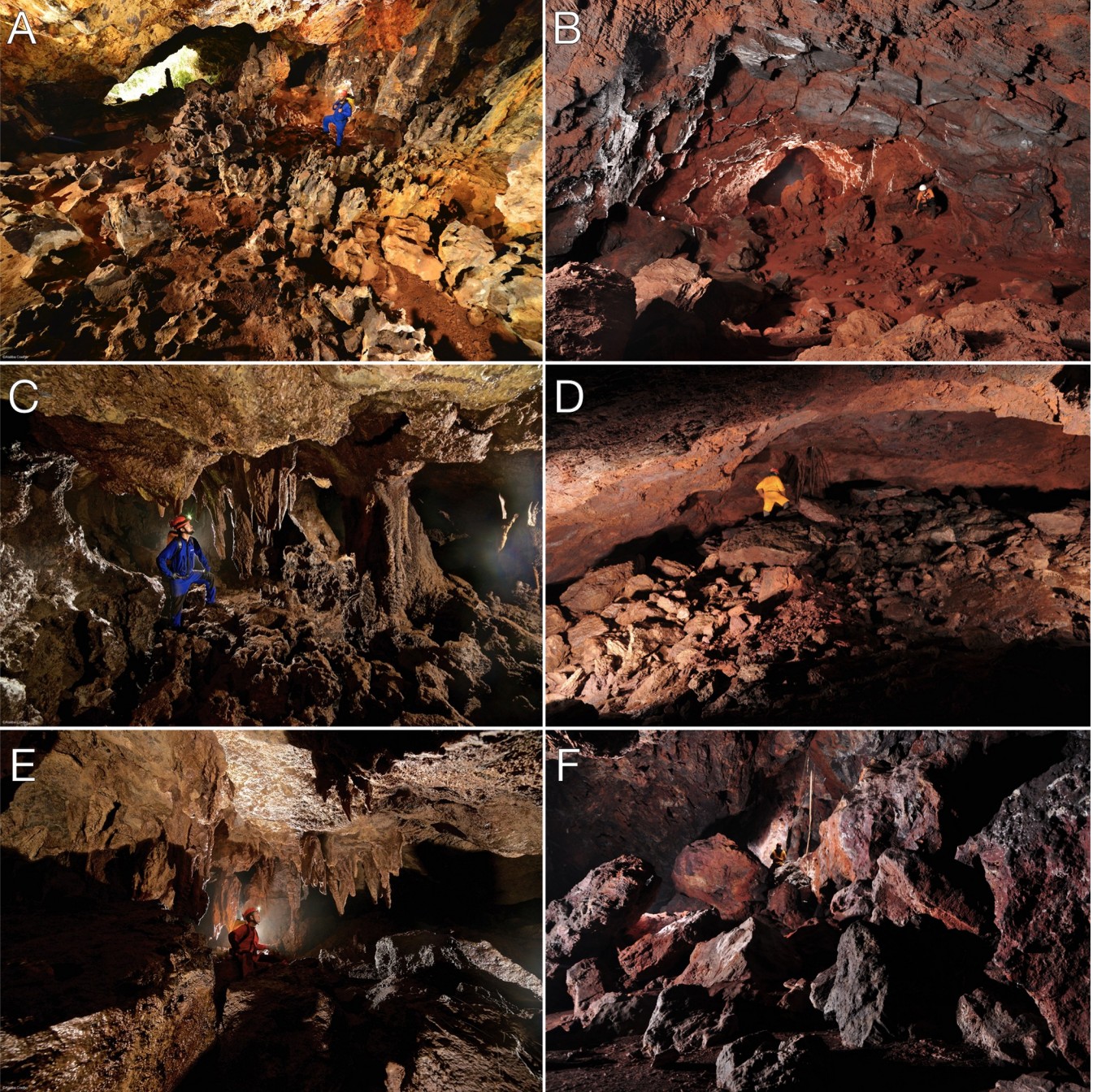

**Fig 2.** Examples of speleothems and corrosion processes in bat caves (A, cave N3-0023; C, cave N4WS-0067; and E, cave S11B-0094) and non-bat caves (B, cave N4E-0008; D, cave N4WS-0015; and F, cave N5SM2-0021). In A and C, deep corrosion processes can be observed in the cave floor. Phosphate stalactites, like in C and E, were very rare and, so far, identified only in active or inactive bat caves. In B, D and E, the roughness of the walls and ceiling is a product of erosive action, not corrosive action. In D and F, the large rebates stand out, which is also an erosive process. (Photo credits: Ataliba Coelho, for A, C and E; Allan Calux, for B, D and F).

In the guano profile N5SM2-0099 there was a small decrease in total organic matter with depth. Overall, organic carbon ranged from 12.2% to 31.2%, values that were recorded in the surface guano. The average % of organic carbon was 21.5%. Nitrogen reached the highest value (7.4%) in the fresh surface guano of sample S11D-0083-Sup, while the lowest value

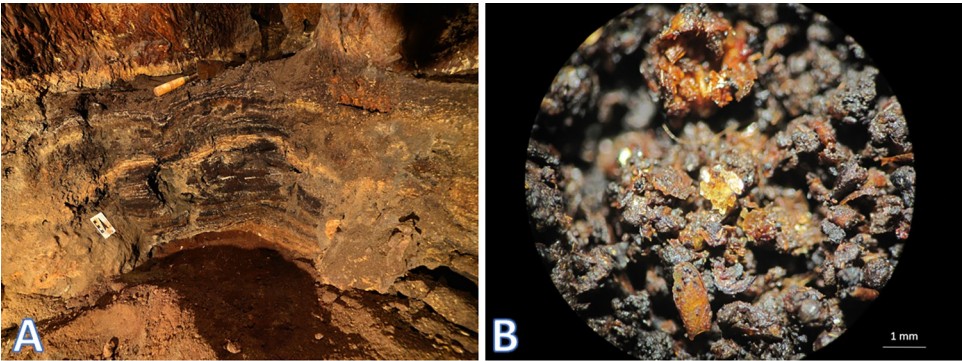

**Fig 3.** A) Profile of guano deposit 65 cm deep at cave N3-0023-P2, in the Carajás National Forest, Brazilian Amazonia. B) Detail of small guano aggregates with remains of insect exoskeletons. Sample from bat cave S11A-0036.

(2.6%) was also obtained on the surface of the profile N3-0023-P2-C1. The values of the C/N ratio varied between 3 and 8 (Table 2).

The guano, rich in organic material, were also rich in $P_2O_5$, which ranged from 1.68%, in the surface sample (N5SM2-0019Sup) to 11.96%, in the deepest sample of profile N3-0023-P1

**Table 2. Chemical profile of guano samples from bat caves in the Carajás National Forest, Brazilian Amazonia.** Depth of sample in centimeters.

| Sample | Depth (cm) | pH | TOM % | OC% | N % | C/N | $P_2O_5$% | Ca % | $K_2O$ % | Mg % | S % | Fe (mg/kg) |
|---|---|---|---|---|---|---|---|---|---|---|---|---|
| N3-0023-P1-C1* | 2 | 4.5 | 46.8 | 24.87 | 3.88 | 6 | 6.32 | 5.21 | 1.16 | 0.30 | 1.16 | 12,773 |
| N3-0023-P1-C2 * | 8 | 4.4 | 41.4 | 21.73 | 3.34 | 7 | 6.00 | 5.01 | 1.40 | 0.17 | 1.58 | 6,956 |
| N3-0023-P1-C3 * | 17 | 4.5 | 48.0 | 25.34 | 3.81 | 7 | 3.72 | 5.20 | 1.46 | 0.23 | 1.38 | 6,120 |
| N3-0023-P1-C4 * | 26.5 | 4.6 | 37.9 | 19.91 | 3.35 | 6 | 10.09 | 7.48 | 1.14 | 0.19 | 1.48 | 14,724 |
| N3-0023-P1-C5 * | 37 | 4.5 | 51.5 | 27.36 | 4.28 | 6 | 2.85 | 3.23 | 1.20 | 0.20 | 1.01 | 6,989 |
| N3-0023-P1-C6 * | 51.5 | 4.6 | 39.3 | 20.64 | 2.84 | 7 | 8.72 | 11.44 | 1.12 | 0.20 | 2.30 | 13,982 |
| N3-0023-P1-C7 * | 62.5 | 4.5 | 46.0 | 24.31 | 4.58 | 5 | 11.96 | 1.91 | 1.08 | 0.21 | 0.23 | 57,405 |
| N3-0023-P2-C1 * | 16.5 | 2.2 | 40 | 20.87 | 2.56 | 8 | 3.38 | 0.51 | 0.16 | 0.04 | 0.05 | 22,942 |
| N3-0023-P2-C2 * | 33.5 | 2.2 | 41.9 | 22.01 | 4.55 | 5 | 4.87 | 0.56 | 0.21 | 0.04 | 0.02 | 33,732 |
| N3-0023-P3-C3 * | 39.5 | 2.1 | 35.3 | 18.38 | 2.97 | 6 | 3.56 | 0.33 | 0.56 | 0.02 | 0.04 | 26,345 |
| N5SM2-0099-C1 * | 7.5 | 5.4 | 46.4 | 24.53 | 5.55 | 4 | 4.76 | 0.67 | 1.25 | 0.58 | 0.03 | 3,131 |
| N5SM2-0099-C2 * | 30 | 4.9 | 42.4 | 22.35 | 5.07 | 4 | 6.08 | 1.20 | 1.47 | 0.97 | 0.54 | 12,526 |
| N5SM2-0099-C3 * | 61.5 | 3.9 | 38.1 | 19.92 | 4.03 | 5 | 3.44 | 3.18 | 1.28 | 0.14 | 1.88 | 9,848 |
| N5SM2-0099-C4 * | 86.5 | 3.5 | 36.8 | 19.29 | 4.03 | 5 | 9.14 | 0.70 | 1.56 | 0.12 | 0.30 | 54,594 |
| S11C-0041-C1 * | 2 | 4.5 | 34.9 | 17.37 | 5.04 | 3 | 1.89 | 0.56 | 1.01 | 0.23 | 0.11 | 1,903 |
| S11C-0041-C2 * | 30 | 4.5 | 37.7 | 19.26 | 5.62 | 3 | 2.59 | 0.74 | 1.05 | 0.24 | 0.17 | 23,564 |
| N5SM2-0019Sup * | 2 | 2.4 | 27.69 | 14.18 | 2.89 | 5 | 1.68 | 0.23 | 0.06 | 0.04 | 0.02 | 29,175 |
| N4WS-0072Sup * | 2 | 4 | 41.58 | 21.94 | 3.68 | 6 | 3.97 | 0.77 | 0.19 | 1.12 | 0.03 | 10,223 |
| S11A-0036Sup * | 2 | 5.2 | 44.1 | 22.57 | 5.26 | 4 | 4.16 | 0.45 | 2.54 | 0.23 | 0.24 | 10,744 |
| S11B-0168Sup | 2 | 3.1 | 24.23 | 12.21 | 2.63 | 5 | 9.08 | 0.4 | 1.09 | 0.09 | 0.04 | 59,280 |
| S11D-0083Sup * | 2 | 5.6 | 45.11 | 23.64 | 7.41 | 3 | 3.14 | 0.67 | 1.58 | 0.38 | 0.14 | 2,445 |
| S11D-0121Sup | 2 | 5.3 | 58.35 | 31.18 | 3.49 | 9 | 4.09 | 1.71 | 2.09 | 0.82 | 1.14 | 23,445 |

TOM: Total organic matter; OC: Organic carbon; N: Nitrogen; C/N: Carbon to nitrogen ratio; $P_2O_5$: Phosphorus pentoxide; Ca: Calcium; $K_2O$: Potassium oxide; Mg: Magnesium; S: Sulphur; Fe: Iron, in milligram per kilogram.

* Denotes active and inactive bat caves, as described in S1 Table.

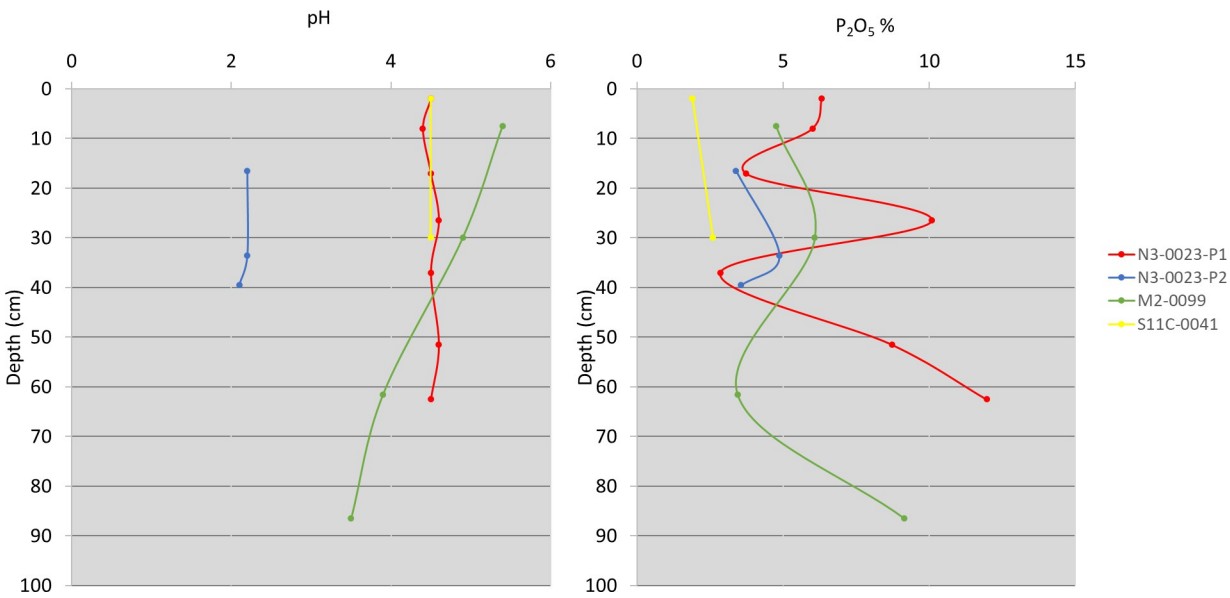

**Fig 4. pH and phosphorus pentoxide variation in samples from different depths taken from guano deposits in three bat caves in Carajás National Forest, Brazilian Amazonia.**

(Fig 4). The presence of calcium (Ca) was high in the profile N3-0023-P1, with values above 11%. $K_2O$ and Mg were also present in guano, with values reaching 2.54 and 1.12%, respectively. Sulfur (S), in turn, reached 2.3% in the N3-0023-P1 profile. The expressive presence of Fe was noteworthy, and reached more than 59,280 mg/kg in a fresh surface sample (S11B-0168) and a lower value of 2,445 mg/kg, also in a fresh surface sample (S11D- 0083Sup).

**pH and water chemistry analysis.** The pH of the circulating waters was complex in the active bat caves (N5SM2-0099, S11A-0036), indicating very acidic values, as well as slightly alkaline values (S3 Table). Outliers occurred in the same cave. An extremely acidic value (pH 1.46) was recorded in a pool with guano in the bat cave N5SM2-0099, with the highest pH value (7.59) in a resurgence with fresh guano in cave S11B-0168, a non-bat cave. The median value was pH 4.9, and the average was 4.7 ± 2.2.

No neutral or alkaline pH values were recorded in currently inactive bat caves (N3-0023, S11B-0094 and N4WS-0067). The values were very acidic and more homogeneous, with a median pH value of 2.9, and average of 3.2 ± 0.9. The most acidic value was 2.2, obtained in a clear water dripping pool in cave N4WS-0067. The least acidic value, 5.9, was recorded in a dark-colored stagnant pool of water caused by guano (S3 Table).

In two caves without the presence of bats or guano (N5S-0011 and N5S-0012), the pH of circulating water varied between 3.81 and 4.7, with a median of 4.0, and average of 4.1 ± 0.3, indicating a pH that is also acidic, but without the extremely acidic values recorded in bat caves. For the surface water samples, pH varied little, from 4.61 to 5.47, with a median value of 5.0, and average of 5.0 ± 0.3, also indicating less acidic values than in bat caves (S3 Table). $NH_4$ concentrations were obtained from just two samples of circulating water (0.15 and 0.46 mg/L —Table 3). Iron concentrations varied from 0.1 up to 4.20 mg/L, while $PO_4$ varied from 0 up to values > 100 mg/L, and $NO_3$ from 8.00 up to > 400 mg/L (Table 3).

**Radiocarbon dating.** [14]C dating of eight guano samples taken from the base of the deposits in six bat caves first identified two of them as dating from the end of the Pleistocene: 22,876 – 22,469 cal BP (Beta 544608) and 18,191 – 17,857 cal BP (Beta 521294) (Table 4). One sample corresponded to the upper Holocene, 7,891–7,878 cal BP (Beta 521292). All the other

**Table 3. Chemical parameters (ammonium, iron, phosphate, and nitrate) of circulating waters samples from bat caves in the Carajás National Forest, Brazilian Amazonia.**

| Cave | Sample | $NH_4$ (mg/L) | Fe (mg/L) | $PO_4$ (mg/L) | $NO_3$ (mg/L) |
|---|---|---|---|---|---|
| N3-0023* | N323-W01 | - | 0.30 | 1.30 | > 400 |
| | N323-W02 | - | 0.10 | 6.40 | 34.40 |
| | N323-W03 | 0.15 | 3.60 | 97.80 | > 400 |
| | N323-W05 | - | 0.40 | 22.10 | 86.00 |
| N4WS-0072* | N4WS72-W01 | - | 0.20 | 5.10 | 15.60 |
| | N4WS72-W02 | - | 2.25 | > 100 | 30.00 |
| | N4WS72-W03 | - | 1.25 | 40.10 | > 400 |
| | N4WS72-W04 | 0.46 | 4.20 | 55.50 | 64.00 |
| N5S-0011 | N5S11-W01 | - | 0.15 | 2.40 | 18.40 |
| | N5S11-W02 | - | - | 1.50 | 15.60 |
| | N5S11-W03 | - | 0.70 | 0.00 | 12.80 |
| N5S-0012 | N5S12-W01 | - | 0.10 | 2.60 | 22.40 |
| | N5S12-W02 | - | 0.25 | 2.60 | 9.60 |
| | N5S12-W03 | - | 1.45 | 1.00 | 8.00 |

* Denotes active and inactive bat caves, as described in S1 Table.

sample dating were for the period of the last 4,000 years: 3,450 – 3,323 cal BP (Beta 544609); 2,464–2,305 cal BP (Beta 521293); 2,183–2,015 cal BP (Beta 521295); 2,018–1,890 cal BP (Beta 544607); and 1,882–1,728 cal BP (Beta 527797). Two ages were determined for a 17 cm-long phosphate stalagmite. The base was dated as 3,892–3,694 cal BP (Beta 534952) and the top as 10,369–10,357 cal BP (Beta 534951).

**Speleothems characterization.** The main speleothems identified in the iron ore caves were coralloids, crusts, flowstones, *pingentes*, stalactites, stalagmites and columns (Figs 2 and 5). Although present in several caves, those speleothems were more frequent, more abundant and more developed only in the active and inactive bat caves (Fig 2).

The coralloids were small, maybe nodular, globular, botryoidal or coral-like in shape and situated mainly on walls, floors and boulders on the cave floor (Fig 5A). Coralloids colors

**Table 4. Estimated ages based on radiocarbon analysis for guano samples and one stalagmite from bat caves in the Carajás National Forest, Brazilian Amazonia.**
Guano samples were obtained from trenches or from a sample hole, all excavated on the cave floor.

| Sample | Source (Depth, in cm) | Conventional age | Calendar calibration (95.4% probability) |
|---|---|---|---|
| N3-0023-P2-SUP* | Trench (1) | 4.850 +/- 30 PB | 5.606 – 5.468 cal BP |
| N3-0023-P2-33* | Trench (33) | 6.560 +/- 30 PB | 7.505 – 7.410 cal BP |
| S11B-0094-23* | Trench (23) | 14.860 +/- 40 BP | 18.191 – 17.857 cal BP |
| S11B-0094-92* | Trench (92) | 18.850 +/- 50 BP | 22.876 – 22.469 cal BP |
| S11B-0094-G0* | Stalagmite bottom (0) | 9.130 +/- 30 BP | 10.296–10.187 cal BP |
| S11B-0094-G17* | Stalagmite top (17) | 3.560 +/- 30 BP | 3.892–3.694 cal BP |
| N5SM2-0099* | Trench (90) | 2.050 +/- 30 BP | 2.018–1.890 cal BP |
| S11C-0041-65* | Trench (65) | 2.180 +/- 30 BP | 2.183–2.015 cal BP |
| N5S-0063-60* | Trench (60) | 2.370 +/- 30 BP | 2.464–2.305 cal BP |
| N4WS-0067* | Hole (5) | 1.910 +/- 30 BP | 1.882–1.728 cal BP |

* Denotes samples from active and inactive bat caves, as described in S1 Table.

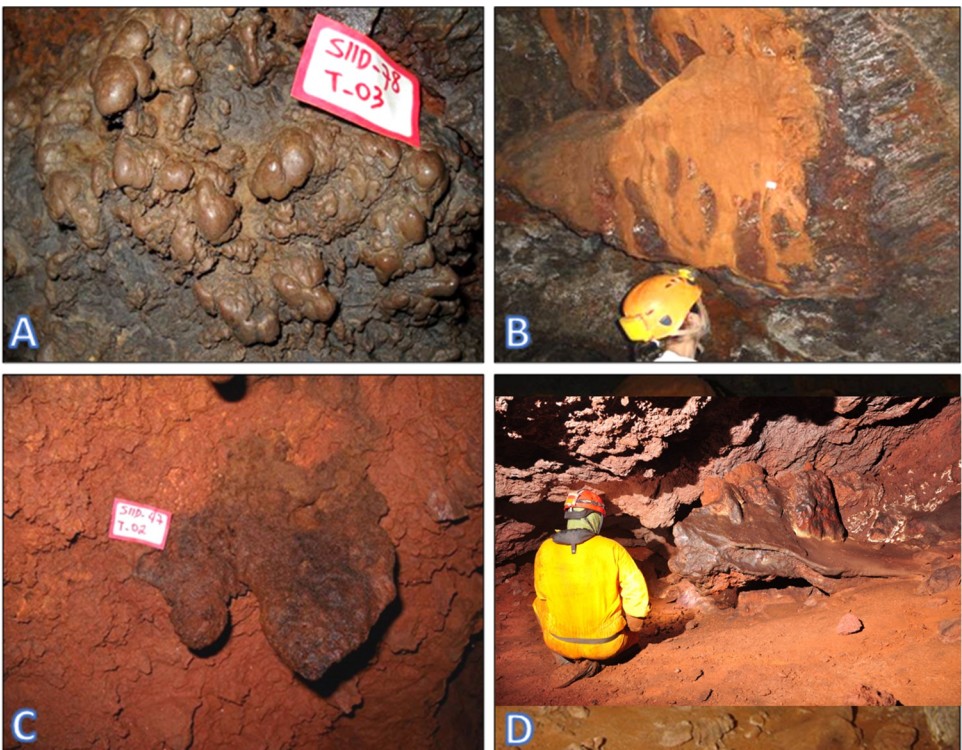

**Fig 5. Examples of speleothems identified in iron ore caves in the Carajás National Forest, Brazilian Amazonia.**
A) Coralloids in cave S11D-0078; B) Flowstones on the wall of cave S11D-0001; C) *Pingente* at the ceiling of cave
S11D-0047; D) Crusts on the floor of cave N4WS-0015. None of those caves are bats caves.

varied from whitish, to red, yellowish, dark brown or light grey, and consisted of iron oxide-hydroxide, especially hematite ($Fe_2O_3$) and goethite ($FeO(OH)$) or of phosphates with a predominance of strengite ($Fe^{3+}(PO_4) \cdot 2H_2O$), leucophosphite ($KFe^{3+}_2(PO_4)_2(OH) \cdot 2H_2O$) and phosphosiderite ($Fe^{3+}(PO_4) \cdot 2H_2O$) (usually associated) and, less often, spheniscidite (($NH_4$) $Fe^{3+}_2(PO_4)_2(OH) \cdot 2H_2O$), taranakite ($K_3Al_5(PO_3OH)_6(PO_4)_2 \cdot 18H_2O$), rodolicoite ($Fe^{3+}(PO_4)$) and hureaulite ($Mn^{2+}_5(PO_3OH)_2(PO_4)_2 \cdot 4H_2O$). Two sulphates, gypsum ($Ca(SO_4) \cdot 2H_2O$) and jarosite ($KFe^{3+}_3(SO_4)_2(OH)_6$) were also detected among the coralloids. Coralloids composed exclusively of iron phosphates usually presented from 36% to 38% of $P_2O_5$, and 39% to 40% of $Fe_2O_3$. Sulphate coralloids may contain up to 13% of $SO_3$.

Crusts (Fig 5D) are relatively frequent in the caves and may cover small areas or, in some cases, practically the entire cave, like in some bat caves in the CNF. They are generated by seeping and flowing water, and mainly cover the floors, but they can also cover walls, ceilings and boulders (but see [37] for a discussion on the crust origins). They may be compact, laminated, banded or micro breccia and can be up to a few centimeters thick. Crusts were red, yellow, white or dark grey, and predominantly consist of iron oxides and hydroxides (hematite and gothite) and phosphates, including strengite ($Fe^{3+}(PO_4) \cdot 2H_2O$), phophosiderite ($Fe^{3+}(PO_4) \cdot 2H_2O$), leucophosphite ($KFe^{3+}_2(PO_4)_2(OH) \cdot 2H_2O$), taranakite $K_3Al_5(PO_3OH)_6(PO_4)_2 \cdot 18H_2O$, Hannayite (($NH_4$)$_2Mg_3(PO_3OH)_4 \cdot 8H_2O$), vashegyite ($Al_{11}(PO4)_9(OH)_6 \cdot 38H_2O$), spheniscidite (($NH_4$)$Fe^{3+}_2(PO_4)_2(OH) \cdot 2H_2O$) and variscite ($Al(PO_4) \cdot 2H_2O$). Crusts formed by sulphates alone (gypsum ($Ca(SO_4) \cdot 2H_2O$), aluminite ($Al_2(SO_4)(OH)_4 \cdot 7H_2O$) and felsőbányaite ($Al_4(SO_4)(OH)_{10} \cdot 4H_2O$) were identified as well as others associated with phosphate (francoanellite $K_3Al_5(PO_3OH)_6(PO_4)_2 \cdot 12H_2O$). Crusts formed exclusively by phosphates

(newberyite, $Mg(PO_3OH)\cdot 3H_2O$ and monetite $Ca(PO_3OH)$)) consisted of 21.7% MgO, 40% $P_2O_5$ and 3.4% CaO.

Other speleothems present in the caves were the flowstones on the walls, originating from solutions coming from fractures and small conduits known as canaliculi. The flowstones may form draperies and small rimstone dams on sloping walls, their colors being predominantly red, yellow and light brown (Fig 5B). The flowstones were mainly composed of oxides and hydroxides of iron or aluminum (goethite, hematite, gibbsite $(Al(OH)_3)$, lepidocrocite $(Fe^{3+}O(OH))$ and bayerite $(Al(OH)3)$). Flowstones composed exclusively of phosphates (associations of phosphosiderite $(Fe^{3+}(PO_4)\cdot 2H_2O)$, sphenisidite $((NH_4)Fe^{3+}_2(PO_4)_2(OH)\cdot 2H_2O)$ and strengite $(Fe^{3+}(PO_4)\cdot 2H_2O)$), were less common. A sample of flowstone composed of iron and aluminum oxides and hydroxides consisted of 80.8% $F_2O_3$ and 2.9% $Al_2O_3$.

The *pingentes* are downward projections from the ceiling or sloping walls consisting of ferruginous material, similar to stalactites, but with no central canal or concentric lamination (Fig 5C). In general, they have many empty spaces inside them. They were either red or reddish-brown and up to 30 cm long in bat caves, but just a few centimeters wide. Analysis of samples indicated a composition of oxides and hydroxides of iron (hematite, goethite and lepidocrocite) and aluminum (gibbsite).

Stalactites and stalagmites were the most surprising forms in the caves and present only in caves with past or current presence of bats (Fig 2). The stalactites had a central canal, concentric growth layers and lengths of up to 1.5 meters. These rare speleothems were usually either dark brown, yellowish or reddish in color. The stalagmites had convex lamination and could take the shape of a cone or a candle. They were less frequent in occurrence than the stalactites and could reach a height of 50 cm. There were also some columns in the caves. The analysis of 18 samples taken from stalactites and stalagmites only identified phosphatic minerals paragenesis: sphenisidite-leucophosphite, strengite-phosphosiderite, leucophosphite-sphenisidite-phosphosiderite and leucophosphite-phosphosiderite-strengite-sphenisidite. In some of the samples, however, only a single phosphatic mineral was identified (leucophosphite or strengite). One iron phosphate stalactite registered 37.5% of $P_2O_5$ and 41.3% of $Fe_2O_3$.

## Discussion

Here we provide evidence of the role of bats as ecosystem engineers in iron ore caves in the Carajás National Forest (CNF), in the Brazilian Amazonia, an area with > 1.500 caves, some holding > 150,000 bats. Acid solutions generated by the decomposition of guano and possible associated microbial activity produced various forms of corrosion in the floor and walls of those bat caves, resulting in their enlargement. In fact, bat caves were deeper, larger and bulkier than the regional average, plus had more abundant, diversified and bigger speleothems when compared with other caves. In an example of bioengineering, we provided the first evidence that the long-term presence of bats (up to 23,000 years before present) and the guano deposits they produce mediated biological and chemical interactions which, in turn, contributed to alter the geomorphology of those iron ore caves.

### The expressive chemical deposits (speleothems) originated from the guano

Most of the speleothems in the iron ore caves in the CNF are relatively small features mainly formed of hematite, goethite and gibbsite, derived from the weathering of the iron-bearing rock with a predominance of crusts and coralloids. In the active or inactive bat caves with deposits of guano, the speleothems are larger more abundant and more diversified. A variety of complex reactions takes place in the guano, especially bacterial decomposition, liberating

nitric, phosphoric and sulphuric acids which react with the rock or the sediments to form secondary minerals [41,42].

Coralloids, crusts, stalagmites and stalactites predominate, largely composed of phosphate, most notably, the minerals leucophosphite, phosphosiderite, strengite and spheniscidite. The phosphatic minerals in the caves we analyzed are derived from the bat guano, given that the banded iron formation of Carajás has very low $P_2O_5$ content (an average of 0.01%—[43]). Sulphate crusts are also present consisting of gypsum, aluminite and felsőbányaite.

In addition to expressive $P_2O_5$ and $Fe_2O_3$ content, the guano in Carajás caves has $K_2O$, CaO, $Na_2O$, and $SO_3$ availability. Aluminum and iron may be made available by the iron-bearing rocks and the authigenic ferruginous clays. Those conditions have made possible the formation of a whole set of diversified phosphate and sulphate minerals, including rare ones like rodolicoite and hureaulite [see 44 for further examples].

## The iron dissolution

Iron oxides are generally compounds with low to very low solubility. In natural systems where most of the iron is in the form of Fe(III) oxides, iron is found particularly in the immobile form. As large amounts of iron circulate in all parts of ecosystems (biota, water and soil), mobilization of iron (dissolution) in oxide reserves can occur [45]. One of the more recently recognized environmental roles of iron is that Fe(III) acts as an electron acceptor in microbial respiration, commonly referred to as dissimilatory reduction of Fe(III) [46]. According to Cornell and Schwertmann [45], the principle behind this process is the potential of Fe(III) in solid Fe oxides to accept electrons from the biogenic dissimilation (oxidation) of organic compounds [see also 37].

Iron is present in the circulating waters inside the caves in the CNF, with the highest values in inactive bat caves N4WS-0072 and N3-0023. This is total iron, i.e., the soluble form cannot be distinguished. Parker et al. [47], however, also working in CNF caves, recorded an extremely low pH of 2.11 in a puddle with guano. This puddle also contained high organic carbon (13 mg L-1) as well as high total iron, of which >74% was in the soluble Fe(II) form. Iron is present in the vast majority of phosphate speleothems, as well as iron oxide-hydroxides (hematite, goethite and lepidocrocite). Fe(II) is clearly being exported out of the caves, where it could be oxidized and reprecipitated with iron oxides. These data show that iron caves, particularly bat caves, are favorable environments for iron(III) solubility. Ammonium was identified in two bat caves. $NH_4$ also forms phosphates, like spheniscidite. Phosphate ($PO_4$) showed the highest values in bat caves, being responsible for the expressive formation of phosphate speleothems in these guano-rich environments. Nitrate was also more expressive in bat caves. However, nitrates are very soluble and quickly leached out.

The reduction of iron in extremely acidic environments is significant since the solubility of ferric ions (FeIII) in acidic media is higher. The solubility of ferric ions is highly dependent on pH, being more available in extremely acidic conditions, with a high redox potential [48,49]. Johnson and McGinness [48] demonstrated the ability of the dissimilatory reduction of ferric ions by *Acidiphilium* SJH under aerobic conditions, in addition to observing that at pH 2 the dissolution of minerals is accelerated, even though the optimal pH for the growth of this acidophilus is around 3. They concluded that contact between the bacteria and the ferric mineral was necessary for reductive dissolution to occur. Several strains of dissimilatory iron-reducing bacteria are being used in several studies, including *Shewanella alga*, *Sh. putrefaciens*, and *Geobacter metallireducens*, among others [45].

The reduction of microbial iron depends on the type of electron donors and acceptors. Fredrickson et al. [50] reported that poorly ordered iron oxides are the main source of iron for

anaerobic microbial reduction because they have a larger surface area and they are more bio-available. However, according to Roden and Urrutia [51] and Hansel et al. [52], crystalline iron oxides can also undergo microbial reduction and, as they are more abundant, they can significantly contribute to the formation of Fe(II). Liu et al. [53] summarized the factors that determine the rate of microbiological reduction of Fe(III): direct contact between organisms and iron oxides; oxide properties, mineral shape, crystallinity and particle size; Fe(II) sorption; biomineralization of Fe(II) and the presence of organic ligands that increase dissolution. The main source of electron donors for iron-reducing microorganisms is organic matter, including simple organic acids, amino acids, sugars, mono-aromatic compounds and long-chain fatty acids [46].

Parker et al. [47] proposed that the iron caves of the Quadrilátero Ferrífero, in Minas Gerais, may be at least partially formed by biospeleogenesis, that is, by the reductive dissolution of Fe(III) phases by reducing bacteria. This microbial activity together with the hydraulic export of soluble biogenic Fe(II) would lead to an increase in the porosity of the strata containing Fe(III) and, finally, to the formation of caves. In the bat caves in the CNF, very favorable conditions for the microbial reduction of Fe(III) occur, including an environment that is often extremely acidic, plus an abundance of organic matter represented by guano. This microbiological mechanism would be responsible for the forms of corrosion on the walls and floors of caves, including flutes, pinnacles, holes and corrosion channels (Fig 6). The hydraulic export of Fe(II) would be controlled mainly by the dripping and the pluvial runoff existing in all the bat cellars. This mechanism would be responsible for the expansion of the caves.

## Bats and biogenic corrosion in caves—Operating mechanisms

Biogenic corrosion associated with the presence of bats and their guano in caves have been previously reported [54–58]. However, this subject has received more attention recently due to a better understanding of the chemical processes involved, the mineralogical evolution of guano, and the role played by bats and their guano on post-speleogenetic cave modification due to condensation-corrosion processes [e.g. 59–63]. Bats and their guano may influence cave development processes in different ways (Fig 7): i) thermally, because the body heat emanated by thousands of individuals and the decomposition of large guano deposits they produce may increase the cave temperature in several degrees; ii) climatically, when the water in bats´ respiration can saturate the ambient air and may condense, covering parts of the cave surface with a film of water; and iii) physicochemically, when the $CO_2$ and urea released by bats´ respiration and urine, respectively, reacts with other chemical compounds in the cave, leading to other physicochemical reactions [64].

Bat caves in general tend to have higher humidity and higher temperatures (~ 34˚C to 40˚C in hot chambers and in the most internal zones) and higher concentrations of ammonia [15]. The high temperatures identified in bat caves have been pointed out to potentialize air convection and condensation of acid-rich corrosive aerosols on cooler walls and ceilings, contributing to the dissolution of the surrounding rock [42,63,65]. In Carajás, another important thermal factor operates: the water percolating the caves is heated when crossing the iron ore layer above.

The guano deposits we studied in Carajás were acidic (pH 2.1 to 5.6) and samples from the base of the deposits were the most acidic, a pattern observed in other caves [66]. Parker et al. [47], recorded an extremely low pH of 2.11 in a puddle with guano in the CNF, and this pool also contained high organic carbon (13 mg.$L^{-1}$) as well as high total iron, of which >74% was in the soluble form. Scherer [67] performed seven pH measurements in circulating water in two bat caves (S11B-0094 and S11D-0083) in the Carajás region. Values ranged from 3.17 to 4.0, with an average pH of 3.59.

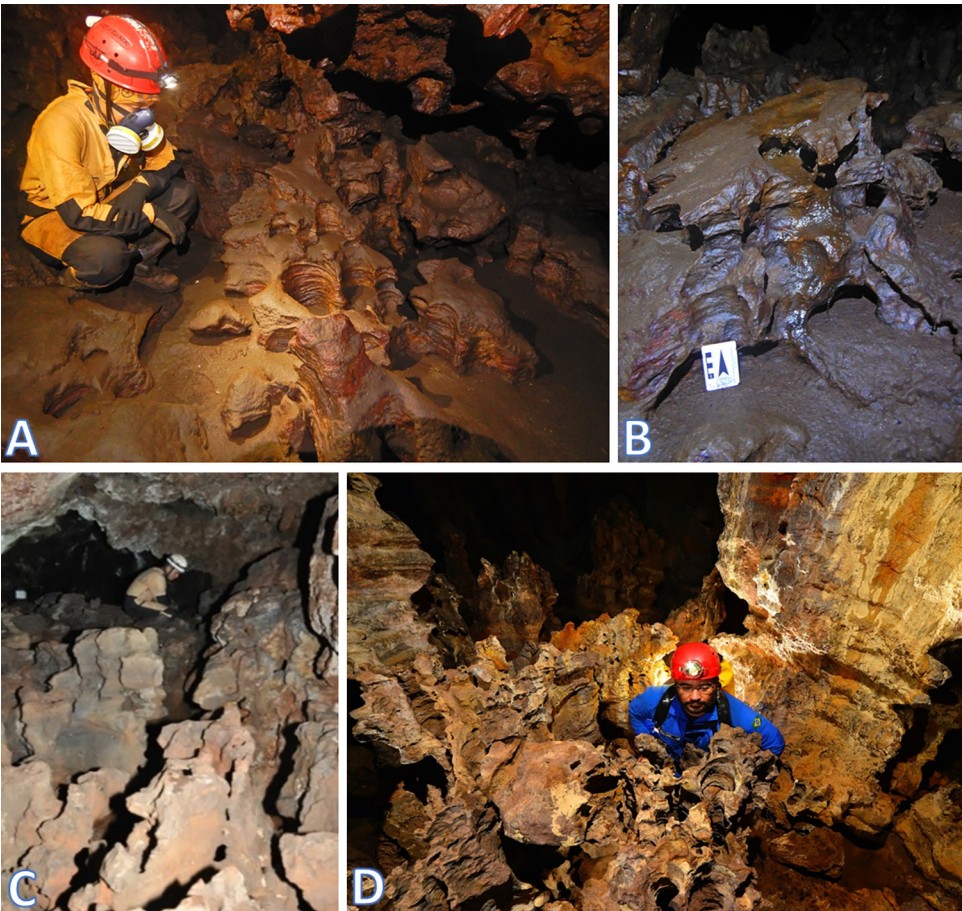

**Fig 6. Examples of corrosion on the floor and walls in bat caves in the Carajás National Forest, Brazilian Amazonia.** A) Dripping holes on the floor covered by bat guano, in the active bat cave N5SM2-0099; B) Corrosion features on fresh guano, in the active bat cave S11A-0036; C) Floor without guano, already washed by water, evidencing pinnacles, in the inactive bat cave N4WS-0067; D) Pinnacles, dripping holes and flutes on the walls, in the inactive bat cave N3-0023.

## Bats as structural engineers

In Carajás, we observed that caves harboring larger guano deposits–both active or inactive–were larger than the regional average, with more abundant, diversified and bigger speleothems. Our observation of a speleothem-rich scenario associated with bat presence is in accordance with previous authors elsewhere. Lundberg and McFarlane [68], studying the Gomantong Caves, in Borneo, identified forms created under-guano deposition (e.g. karren, notches, and arches) and subaerial forms arising from upward thermal convection movements and condensation-corrosion activity (e.g. wall channels, flutes, pockets, cupolas, bell holes). Audra et al. [59], in a complete summary on the role of bats in the late morphologic evolution of caves, distinguished four types of bat-associated forms: 1) those related to crypto-corrosion under guano (e.g. pinnacles, pedestals, karren); 2) those created by condensation-corrosion phenomena (e.g. smoothed walls and ceilings, wall and pillar niches and notches, channels); 3) different deposits and mineralization associated with bat droppings (excreta and urine marks and trickles, phosphate and apatite crusts); and 4) forms related to bat roosting (e.g. bell holes and cupolas).

Complementary, Dandurand et al. [63], working in the Drotsky's Cave, in Botswana, proposed a typological classification of bat-associated speleothems composed by three classes:

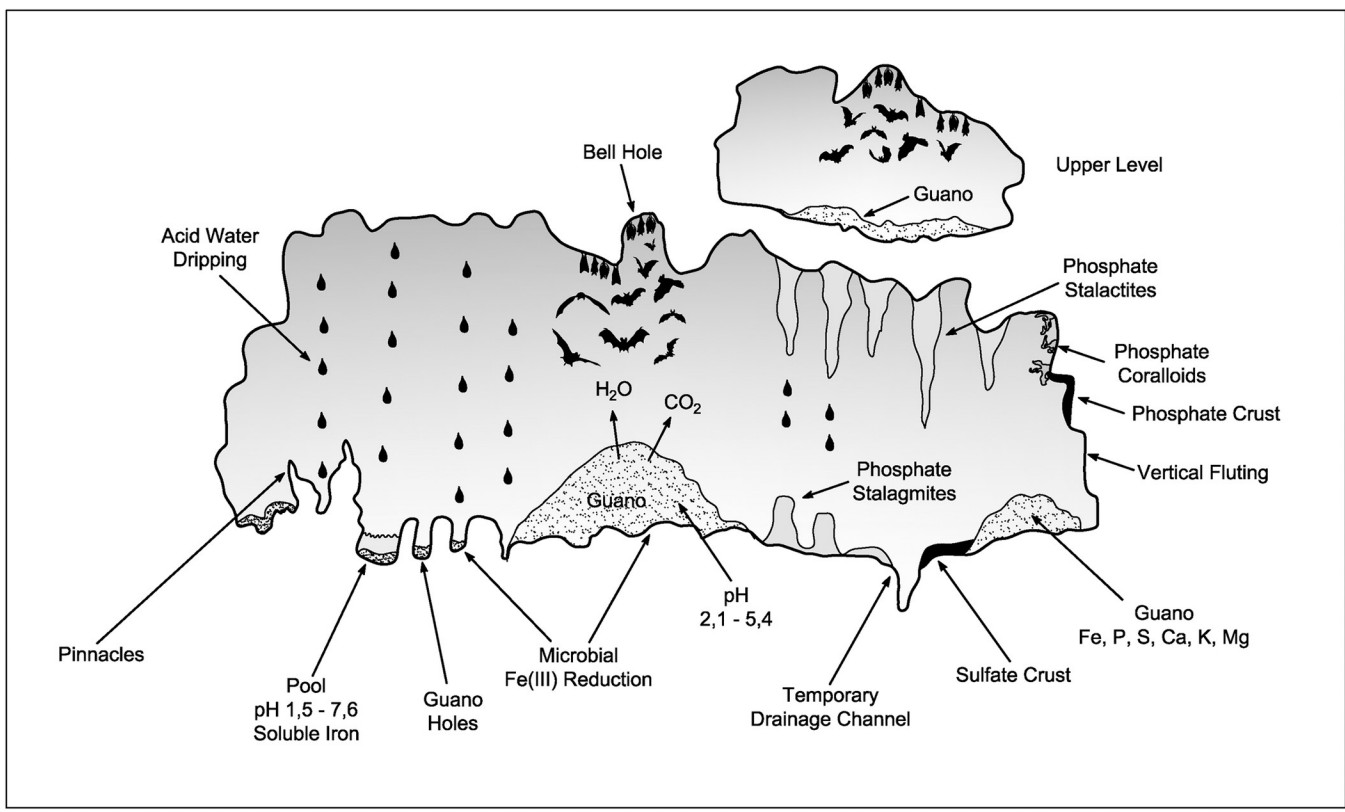

**Fig 7. A variety of complex reactions takes place in the guano, especially bacterial decomposition, liberating phosphoric and sulphuric acids which react with rock or the sediments to form speleothems (stalactites, stalagmites, crusts and coralloids) of phosphates, with a predominance of leucophosphite, phosphosiderite, strengite and spheniscidite, and sulfates (gypsum).** In bat caves, conditions are very favorable for the microbial reduction of iron(III), including an environment that is often extremely acidic and with an abundance of organic matter represented by guano. This microbiological mechanism would be the main responsible for the dissolution of the iron rock. Important values of total iron, including soluble iron, were recorded in pools containing guano. The dripping and temporary drainage processes would be responsible for the elaboration of the forms of corrosion on the walls and floors of bat caves, including pinnacles, drainage channel, vertical fluting and guano holes. Fresh guano deposits also release $CO_2$ and water vapor, which rise through the convection process of hot air produced by the exothermic decomposition of guano. These gases can produce carbonic acid which eventually corrodes the altered host rock and speleothems of phosphates and sulfates. Figure based on Onac and Forti [41].

*Direct microforms* (e.g. bell holes, cupolas, and ceiling alveoli) that develop mainly under the direct influence of roosting bats and where very localized (microscale) corrosion causes faster, more concentrated dissolution; *Indirect mesoforms*, when forms at a smaller scale result indirectly from the presence of bats and reflect the accumulation and mineralization of guano on the ground and its lateral extension; and *Extended macroforms* (e.g. smooth surfaces or highly altered walls, niches and notches, and hemispherical cupolas) result from convection-condensation corrosion processes affecting the entire cavity. In Carajás, we detected coralloids, crusts, flowstones, *pingentes*, stalactites, stalagmites and columns, indicating that the processes described by both Audra et al. [59] and Dandurand et al. [63] are present and help to explain the speleothem-rich scenario observed in those bat caves.

## Long-term effects of bat presence in caves

Bioengineering processes and effects can occur across a variety of temporal and spatial scales. As pointed out by Hastings et al. [3], the structural modifications caused by some engineering organisms to their environment may be large and persist on time scales well beyond than the organism´s individual lifetimes. Improving the knowledge of long-term engineering effects are

among the key gaps to be filled regarding the understanding of engineer fauna in terrestrial habitats, and, in fact, very few studies track the long-term effects or impacts of interactions involving bioengineer species [4]. Corrosion rates in caves can strongly vary and depend on several factors. However, bats were already perceived as significant driver in cave corrosion [54,56,57,59,68]. But the effects of long-term presence of bats in cave weathering and corrosion processes is still poorly described.

In the Gomatong limestone cave system, in eastern Sabah, Malaysia–currently harboring ~600,000 *Chaerephon plicata* bats–biogenic corrosion caused by bats, swifts and their guano was pointed as the main post-speleogenetic driver, responsible for as much as 70–95% of the total volume of the modern cave, over a timescale of 1–2 Ma. [68]. There, bat-related biogenic corrosion was estimated at rates of up to ~34 mm/ka (or 1 m/~30 ka). Previously, in a study on bell holes (blind vertical cylindrical cavities in cave roofs) in the Runaway Bay Caves, Jamaica, Lundberg & McFarlane [69] estimated that a typical bell hole 1 m deep may be formed in some 50 ka. In Drotsky's Cave, a carbonate cave in Northeastern Botswana, the presence of a colony of 30,000–90,000 bats of three species (*Hipposideros vittatus*, *Nycteris thebaica*, and *Rhinolophus denti*) over millennia was pointed as crucial for explaining the cave speleogenetic evolution, with bats being responsible for direct (bell holes, cupolas, ceiling alveoli) and indirect (notches, niches and arches, karren, cupolas and pockets) micro and mesoforms, plus extended macroforms (smooth surfaces, highly altered walls, wall niches and notches, subvertical channels, and hemispherical cupolas) [63]. Moreover, in that cave it is conceivable that some stalagmites, pillars, and stalagmitic floors buried under several meters of guano have completely disappeared as a result of corrosion due to the acidity of the deposit. In the Azé limestone cave, in France, where *Myotis daubentonii* bats are present for more than 54 ka, a calcite flowstone 22 ka ago isolated part of the cave, preventing access and entry of bats in part of that cave [64]. That allowed researchers to better understand the role bats and their guano had on the cave´s morphological changes: in the area inaccessible to bats, cave walls were hardly altered, whereas the parts available for bats experienced the formation of cupolas on the ceiling, which resulted in an upward expansion of the cave's volume, with an estimated retreat of the walls and speleothems of ~3 to 7 mm/ka [64].

In our study in Carajás, radiocarbon dating of guano samples indicates ages ranging from the end of the Pleistocene (22.0 cal kyr BP) testifying to the long duration of the presence of bat colonies in the caves and the effects of such presence. None of the guano deposits we identified were thicker than 1 m, and water activity (dripping and small flows) in most of the bat caves may have eroded or transported those deposits out of the caves. Older guano deposits are likely to occur in the Carajás area, however their identification depends on further prospection. Our radiocarbon dating also indicated that guano trapped in a 17 cm long phosphate stalagmite was dated to 10.2 cal kyr BP (bottom) and 3.8 cal kyr BP (top), giving us a reference for the growth rate of these rare formations. Not considering possible interruptions in the dripping of phosphate rich water and other variables such as climate, then the growth rate of that phosphatic stalagmite was 0.026 mm/year. Our data add evidence on the role that long-term presence of bat colonies may have as a driver for cave speleogenesis and, as far as we know, this is the first report of evidence for non-carbonatic caves worldwide.

## Conservation implications

Ecosystem engineer species are frequently pointed out as having significant impacts upon the physical structure of their habitats and the organisms that live in them [4,5,70,71]. Consequently, ecosystem engineers are also recognized as having high conservation value [6], with some authors proposing those engineer species could speed ecological recovery processes and

easy the re-establishment of endangered species, and should be considered to maximize conservation outcomes [72–74].

Ecosystem engineer species can be affected by naturally changing climates and habitats, but they may be particularly affected by human activity [75,76]. This is the case of the bat caves in our study area. Carajás is the largest and the most important mineral province in Brazil, harboring some of the largest and purest iron ore stocks in the world, plus significative stocks of copper, gold and manganese. The area is operated by Vale S.A., currently the world´s largest iron ore mining company. Vale S.A. also operates a newer mine, known as Serra Sul or S11D, ~50 km south of Carajás. Mineral prospection started in Carajás in the 1960s, but the whole area is under control of Vale S.A. since then. Part of that mineral extraction area was formally protected by the Carajás National Forest (hereafter CNF, IUCN category VI) just in 1998. The CNF covered ca. 412,000 hectares with ~25% of that area devoted to the mineral extraction. In 2017, as part of the environmental compensation for the new S11D mine, ~60,000 ha of the CNF closer to Serra Sul were dismembered and joined to another 19,000 ha to form the Campos Ferruginosos National Park, a more restrictive federal protected area (IUCN category II). Nearly 1,500 caves are known in both Carajás and Serra Sul areas, forming one of the most important speleological areas in Brazil.

According to the Brazilian legislation, all caves in the country belong to the federal government. Until 2008, all the natural caves in Brazil should be treated as a national cultural heritage and should be preserved. In 2008, a new Presidential Decree #6640 stated that caves in areas under commercial activity, like mining, should pass a classification process during the environmental licensing, classifying those caves as having maximum, high, medium or low relevance [77]. Only those classified as having maximum relevance would be fully protected *a priori*. Bat caves met the requirements to be classified as having maximum relevance and, therefore, should be fully protected.

However, the growing world demand for iron ore has led Vale S.A. to request the Brazilian government to expand its mining activities beyond the current limits of extraction in both the Carajás and Serra Sul regions. Vale S.A. then filed requests for environmental licensing in areas four times larger than the current ones, including pristine areas in the protected areas in Serra Sul which are currently off-limits for mineral extraction. Shortly after, in January 2019, Brazil´s Ministry of Mines and Energy unilaterally indicated that it would change and relax the rules for the protection of Brazilian caves in the environmental licensing processes. In September 2020, MME published its Mining Plan 2020–2023, with clear goals to change the legislation for the protection of caves in the country [78]. In fact, in January 2022, a new Presidential Decree (#10935) effectively changed the protection of caves in the country, allowing caves of maximum relevance–like bat caves–to suffer permanent negative impacts, including total destruction [79].

The current scenario of expansion of mineral activities, together with the loosening of licensing and cave protection rules, is a real threat to the conservation of bat caves in the Carajás region. For the bat species that form the large colonies observed (*Pteronotus gymnonotus* and *P. personatus*), the microclimatic conditions in the caves have a direct influence on postnatal development and offspring recruitment [39,80], as those species in general do not tolerate low temperatures due to their lower rates of basal metabolism [81–84]. Thus, bat caves and their high temperatures are fundamental and irreplaceable for the reproduction and maintenance of their populations.

As demonstrated here, bat caves are rare and exceptional caves, the result of very unique ecological, evolutionary and geomorphological processes that have expanded over millennia. In the case of Carajás, nearly 10 bat caves were identified among ~1500 known caves in the area, stressing how unique those caves are. Parts of the processes operating in those caves

points to intricated interactions between biological, chemical and physical associations, which are just beginning to be better understood by science. Allowing the destruction of those caves would represent and unacceptable loss of both geomorphological and biological heritage. Therefore, we urge that such caves and their bat colonies must be fully protected in the Carajás areas, leaving them off-limits of mineral extraction.

## Supporting information

**S1 Fig. Length and volume of caves in Carajás.** Scatterplot graph considering the length (in meters) and volume (in cubic meters) of 1,309 caves in the Carajás National Forest, Pará State, Brazilian Amazonia. Solid diamonds represent active and inactive bat caves, which tend to be longer and bulkier than the majority of the regional caves. Cave size alone does not explain bat choice for these specific bat caves, and the deep corrosion signs and the higher richness and variety of speleothems were observed only in bat caves.
(DOCX)

**S1 Table. Active and inactive bat caves studied in the Carajás National Forest, Pará State, Brazilian Amazonia.** Coordinates, dimensions (length, area and volume), and information on their biogenic morphology and speleothems for 10 caves observed.
(DOCX)

**S2 Table. Caves studied in the Carajás National Forest, Pará State, Brazilian Amazonia.** Cave coordinates and description of samples collected and analyses performed. * Denotes active and inactive bat caves, as described in S1 Table.
(DOCX)

**S3 Table. pH of superficial and circulating water samples collected in caves in the Carajás National Forest, Pará State, Brazilian Amazonia.** * Denotes samples from active and inactive bat caves, as described in S1 Table; ** denotes samples from other caves; and SP denotes samples from superficial waters, including artificial lakes and drainages.
(DOCX)

**S1 File. Laboratory reports with chemistry data for organic matter in guano samples.** Reports issued by the Laboratório de Fertilizantes, Corretivos e Resíduos Orgânicos of the Escola Superior de Agricultura Luiz de Queiroz (ESALQ/USP) indicating organic matter and chemical composition of guano samples analyzed.
(PDF)

**S2 File. Laboratory reports with X-ray diffractometry (DRX) data for guano samples.** Reports issued by the Laboratório de Caracterização Tecnológica, Departamento de Engenharia de Minas e de Petróleo at the University of São Paulo´s Escola Politécnica, indicating mineral identification of guano samples using the powder method and a Panalytical X-ray diffractometer.
(PDF)

**S3 File. Laboratory reports with X-ray diffractometry (DRX) data for speleothem samples.** Reports issued by the Laboratório de Caracterização Tecnológica, Departamento de Engenharia de Minas e de Petróleo at the University of São Paulo´s Escola Politécnica, indicating mineral identification of spelothems samples using the powder method and a Panalytical X-ray diffractometer.
(PDF)

**S4 File. Laboratory reports with X-ray fluorescence spectrometry (FRX) data for guano and speleothem samples.** Reports issued by the Laboratório de Caracterização Tecnológica, Departamento de Engenharia de Minas e de Petróleo at the University of São Paulo´s Escola Politécnica, indicating mineral percentage in samples using the X-ray fluorescence spectrometer.
(PDF)

**S5 File. Laboratory reports with optical emission spectrometry data for guano, soil, and speleothem samples.** Reports issued by the Laboratório de Caracterização Tecnológica, Departamento de Engenharia de Minas e de Petróleo at the University of São Paulo´s Escola Politécnica, indicating Fe, Ni, P, Rb and Zi concentrations (mg/kg) in guano, soil and speleothems samples using the optical emission spectrometer.
(PDF)

**S6 File. Laboratory reports with optical emission spectrometry data for stalactite samples.** Reports issued by the Laboratório de Caracterização Tecnológica, Departamento de Engenharia de Minas e de Petróleo at the University of São Paulo´s Escola Politécnica, indicating Fe, Ni, P, Rb and Zi concentrations (mg/kg) in stalactites samples using the optical emission spectrometer.
(PDF)

## Acknowledgments

We gratefully thank Jocy Cruz, Iuri Brandi, Daniela Silva, Ataliba Coelho, Thadeu Pietrobon, Bruno Scherer, Airton Barata, Narjara Pimentel, Diego Bento, Flávio Ramos and Francisco Cruz Junior for collaboration with the Bat Caves Project: TCCE–ICMBio/Vale N. 1/2018. This manuscript is derived from a postdoc by L. Piló at the Programa de Pós-Graduação em Biologia Animal PPGBA/UFPE, and we thank that program for all the support. E. Bernard has a fellowship from CNPq. This article is dedicated to the memory of Luís Piló, who unfortunately died after its original submission for revision and publication.

## Author Contributions

**Conceptualization:** Luis B. Piló.

**Data curation:** Luis B. Piló, Allan Calux.

**Formal analysis:** Luis B. Piló, Allan Calux, Rafael Scherer.

**Funding acquisition:** Luis B. Piló.

**Investigation:** Luis B. Piló, Allan Calux, Rafael Scherer, Enrico Bernard.

**Methodology:** Luis B. Piló, Allan Calux.

**Project administration:** Luis B. Piló.

**Resources:** Luis B. Piló, Allan Calux.

**Writing – original draft:** Luis B. Piló, Allan Calux, Enrico Bernard.

**Writing – review & editing:** Luis B. Piló, Allan Calux, Enrico Bernard.

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
