## [Decision Letter · Decision Letter 0]

30 May 2022

PONE-D-22-11101Bats as ecosystem engineers in iron ore caves in the Carajás National Forest, Brazilian AmazoniaPLOS ONE

Dear Dr. BERNARD,

Thank you for submitting your manuscript to PLOS ONE. After careful consideration, we feel that it has merit but does not fully meet PLOS ONE’s publication criteria as it currently stands. Therefore, we invite you to submit a revised version of the manuscript that addresses the points raised during the review process.

We look forward to receiving your revised manuscript.

Kind regards,

Il Won Kim

Academic Editor

PLOS ONE

Journal Requirements:

3. We note that Figures 3 and 6 includes an image of a participant in the study. 

[Authors declare no competing interest.]

6. PLOS requires an ORCID iD for the corresponding author in Editorial Manager on papers submitted after December 6th, 2016. Please ensure that you have an ORCID iD and that it is validated in Editorial Manager. To do this, go to ‘Update my Information’ (in the upper left-hand corner of the main menu), and click on the Fetch/Validate link next to the ORCID field. This will take you to the ORCID site and allow you to create a new iD or authenticate a pre-existing iD in Editorial Manager. Please see the following video for instructions on linking an ORCID iD to your Editorial Manager account: https://www.youtube.com/watch?v=_xcclfuvtxQ.

7. We note that Figures 1 and 7 in your submission contain map/satellite images which may be copyrighted. All PLOS content is published under the Creative Commons Attribution License (CC BY 4.0), which means that the manuscript, images, and Supporting Information files will be freely available online, and any third party is permitted to access, download, copy, distribute, and use these materials in any way, even commercially, with proper attribution. For these reasons, we cannot publish previously copyrighted maps or satellite images created using proprietary data, such as Google software (Google Maps, Street View, and Earth). For more information, see our copyright guidelines: http://journals.plos.org/plosone/s/licenses-and-copyright.

a) You may seek permission from the original copyright holder of Figure(s) [#] to publish the content specifically under the CC BY 4.0 license.  

B) If you are unable to obtain permission from the original copyright holder to publish these figures under the CC BY 4.0 license or if the copyright holder’s requirements are incompatible with the CC BY 4.0 license, please either i) remove the figure or ii) supply a replacement figure that complies with the CC BY 4.0 license. Please check copyright information on all replacement figures and update the figure caption with source information. If applicable, please specify in the figure caption text when a figure is similar but not identical to the original image and is therefore for illustrative purposes only.

Additional Editor Comments:

Your manuscript was reviewed by two experts in the field. The review results are quite contradictory, but some of the concerns are shared by both reviewers. If you could present point-by-point responses to the reviewers’ comments (especially Reviewer #2) and revise your manuscript accordingly, I am willing to send out the revised manuscript for the second round of reviews.

Reviewers' comments:

Reviewer's Responses to Questions

**Comments to the Author**

1. Is the manuscript technically sound, and do the data support the conclusions?

Reviewer #1: Yes

Reviewer #2: No

2. Has the statistical analysis been performed appropriately and rigorously? 

Reviewer #1: N/A

Reviewer #2: No

3. Have the authors made all data underlying the findings in their manuscript fully available?

Reviewer #1: Yes

Reviewer #2: Yes

4. Is the manuscript presented in an intelligible fashion and written in standard English?

Reviewer #1: Yes

Reviewer #2: Yes

5. Review Comments to the Author

Reviewer #1: This paper is dedicated of the impact of biocorrosion caused by bats in caves. This topic is very recent but major regarding caves post-speleogenic evolution. For a very long time, this impact was not understood and it led to misunderstanding about some caves features and reshaped speleothems. Actually, researchers are discovering this process all over the world and it is becoming a fundamental aspect that need to be taken in account for cave studies.

The paper submitted to Plos One will contribute to the structuration of this new paradigm and will become a reference for the next researches. It is of high impact, well-constructed, well written and supported by many field and analyses data. Furthermore, the impact of biocorrosion is here studied in a very particular type of caves, formed in iron ore cave, which was never demonstrated before. Apart from purely scientific aspect, this paper shows the scientific value and the interest of these caves which are threatened by the expansion of mining activities. It could contribute to save these caves, the biological heritage they represent but also the bats colonies themselves which are endangered in so many places around the world.

I recommend to publish this paper with minor corrections. Two small corrections are reported in the PDF attached.

One point, which is highlighted in the abstract and in the results, is missing according to me in the discussion is about the size of the caves. You wrote:” Caves currently harboring bat colonies and those with signs of past presence of such colonies had, on average, horizontal projections 4.5 times larger, areas 4.4 times larger, and volumes 5.0 times bigger than the reginal average ». This is a very interesting point but it is important to discuss it. Some people could ask if the bats didn’t choose the largest caves and you have to address this point. Of course, I understood that you are demonstrating how much the caves were enlarged thank to the biocorrosion and I agree. But to reinforce this point, it is needed to come back on these rates and discuss them here in the “Long-term effects of bat presence in caves » chapter. It can even give you some information, based on the erosion rate due to biocorrosion or for how long the caves were occupied by bats for example. Just an estimation which can bring you further than 22ky...

Reviewer #2: See the three attached files:

1. Text with notes

2. Supplementary Material with notes

4. Review

(For the Text with Notes file, I copied the text into Word because it is easier to add comments in that format)

6. PLOS authors have the option to publish the peer review history of their article (what does this mean?). If published, this will include your full peer review and any attached files.

Reviewer #1: **Yes: **BRUXELLES Laurent

Reviewer #2: No

---

## [Author Response · Author response to Decision Letter 0]

10 Aug 2022

Dear Editor, 

 Please find attached our revised version of the manuscript PONE-D-22-11101 “Bats as ecosystem engineers in iron ore caves in the Carajás National Forest, Brazilian Amazonia”. We would like to acknowledge the careful revision provided by both reviewers, which for sure improved the overall quality of our manuscript. Below you will find a detailed list of all the corrections made, with the respective answers to the reviewers. We started by answering Reviewer 2, since his/her comments were more extensive and required further corrections. Moreover, the issue raised by Reviewer 1 was also addressed in our answers to Reviewer 2. Please accept our revised version and let me know if any other information is necessary. 

 Best,

Enrico Bernard

Reviewer 2

Comments from the document “PONE-D-22-11101 review – Copy.pdf”

R2: This might be considered to be ancillary, but I was very disturbed by the apparently wanton destructive sampling of so many speleothems. For example, line 227-230 notes the rare occurrence of stalactites and stalagmites … yet these researchers took 18 of them. The total number of speleothem samples they report (line136) is 108. This is an unbelievable number of samples of a non-renewable resource.

1.>> Speleothems, including stalactites, stalagmites and crusts, were not collected in an unplanned, excessive or aleatory approach. Speleothems were collected from specific caves and, whenever possible, those speleothems collected provided multiple samples used for different purposes, including chemical analysis, radiocarbon dating and the preparation of cutting profiles for stratigraphic analyses. We have reorganized Table S2, detailing the caves sampled and the methodology approach in each of them. A total of 18 stalactites, stalagmites and crusts were collected, producing 108 samples for analysis. So, the 108 samples referred by the reviewer is in fact the total number of samples we analyzed, not speleothems per se. We have corrected the phrase in the text (now along L159-162) for clarification. Several of the samples we analyzed provided first data for Brazil, justifying their collections. Moreover, all collections were strictly authorized by CECAV/ICMBio, the federal institution responsible for cave research and protection in Brazil. 

R2: The very basis of the research – the designation of “bat cave” versus “non-bat cave” is not rigorous at all … in fact does not seem to have been clearly defined at all. How many bats is required to fit the requirement? One bat? one thousand? 

2.>> There is no such a universally accepted definition of how to classify a bat cave, neither there is a threshold number of bats above which a cave could be labelled as a bat cave. Basically, a bat cave is a cave holding an exceptionally large bat population, usually much bigger than the regional average. We improved our definition in the text (L72-75), provided a better description of those caves in Tables S1, and included now one extra reference (Ferreira 2019, reference number 17) which address this problem and possible definitions. 

R2: We are simply told that nine bat caves were chosen (and it is only by looking at the Supplementary text that we discover that two of them had no guano – at least I think that is what Table S1 says– depends on the meaning of “expressive”)

3.>> We have improved the description of our methodology and fixed Table S1 and Table S2, providing clearer information on active and inactive bat caves sampled, their dimensions and characteristics. Moreover, in all the tables bat caves are now clearly assigned.

R2: The research aim is not clearly laid out. Their aim is “to provide evidence of the role of bats as ecosystem engineers” … but they offer no concrete research questions. I soon understood that it is not the bats themselves that effect the engineering, but rather it is the guano. However, I had to surmise this from their immediate jump to “We analyzed guano …”. They do not state, anywhere in the introduction, the very obvious step that they would compare the features of non-bat caves to bat caves (i.e., test the hypothesis that bat caves would show significant differences in process and geomorphology (erosional and depositional features) from non-bat caves. We eventually get to this at line 131-135. I would like the basic research strategy to be explained before the methods section.

4.>> We have improved our rationale in the introduction, addressing our hypothesis and briefly introducing the steps we followed (now along L84-94), which are now better described in the Methodology. 

R2: As I understand it, in order to prove that bats engineer their ecosystem, the authors need to prove that there are significant differences between caves with bats versus caves without bats. Normally, when comparing two populations to see if one shows a significant difference from the other, I would expect samples to be relatively well matched so that, as far as is possible, only one variable is different between the two. In this case, I would expect caves to be selectively matched for as many geographic/geologic variables as possible, with bat presence the only major difference. So, I would expect to see roughly similar sample sizes, and some explanation of the sampling strategy. However, the sampling is astonishingly lopsided: for the detailed studies these authors compared nine bat caves (of which two appear to have no guano, and of which at least two are designated as active and at least one as inactive – I have not found details for all of them) with 26 non-bat caves (very few details/results given for these caves) and for the cave size comparisons they compare the nine bat caves against more than a thousand non-bat caves. I see no effort to control for other (biological or non-biological) variables that might explain differences. Even something very simple as location in relation to scarp face will govern the size of these caves regardless of bat population (smaller further from scarp). Were the nine the only caves in the region that housed bats? … and, as mentioned above, what constitutes a bat cave? also, do we know that all 1,303 non-bat caves were actually devoid of bats? In this geological situation, the “caves” are simply the porosity voids that are big enough for humans to enter … so, I would be surprised if these colonial bats would choose to live in the tiniest ones. The authors do lean rather a lot on the difference in size of the caves. However, I am not sure that comparing size is valid. With reference to comparing size of bat and non-bat caves, I am bothered by the possibility/probability that bat caves are bigger simply because bats choose to colonize only the bigger caves (a question with no clear consensus in bat biology research). These types of colonial bats require a high temperature and humidity to thrive, and small caves inherently have larger surface area (i.e., more energy conducting rock surface) to volume ratios, so it requires more energy to maintain the high temperatures in smaller colonies and in smaller voids. I suggest that, in finding 1303 non-bat caves to compare with only 9 bat caves they are of course going to be sampling many tiny caves, and thus biasing the results towards smaller non-bat caves.

5.>> We really would like to have had similar sample sizes (active bat caves x inactive bat caves x non-bat caves) to test our hypotheses in a statistically flawless approach, controlling all possible variables. However, in the real world this was not possible: As we demonstrated, bat caves are unique and very rare caves. As far as we know, there are only five active bat caves out of ca. 1,500 known caves in Carajás National Forest, and Carajás is one of the best inventoried speleological provinces in Brazil. We sampled three out of those five active bat caves. If active bat caves were more common there, we would be aware of them. The essence of our study was to show exactly that we are facing a very unique and rare process, observed in just 0.2% of a very large sample size (1,309 caves for which we have available data). Contrary to Reviewer 2´s indication, cave size alone does not explain bat choice for these specific caves. If that were the case, other large caves present in the region – and there are other caves as large as the largest bat cave studied [see the new Figure S1] – would also be occupied or show signs of previous occupation by bats. But this was not observed. Only 10 caves (three active and seven inactive bat caves) show signs of past or present occupation by large bat populations and the deep corrosion signs, the richness and variety of speleothems described were observed in only these ten caves. Speleothems in caves with no expressive guano deposits were of local occurrence and small, while the greater diversity, abundance and size of speleothems was found in the bat caves, with stalactites, stalagmites, columns, coralloids and crusts sometimes almost entirely covering the floor, walls and ceilings. This is now clearly presented along L214-218 and L311-314, and illustrated with new panels and a new legend in Figure 2 . Once again, this reinforces how unique and singular this process was, precluding us to have similar standardized and heterogeneous sample sizes.

R2: Manuscript structure is not always quite logical: e.g., In the section “Cave and speleothem characterization” the small-scale features are described and the component minerals listed, starting with all the depositional forms. There is no indication that bat and non-bat caves were being compared until line 236. Then they suddenly offer a comparison of the two types – but from the cave survey data rather than the features in the cave. My obvious question is: why offer all those lists of minerals in the speleothems but fail to report on the comparison between cave types.

 6.>> We have now reorganized the sequence our results are presented, starting with “Cave dimensions” and moving the “Speleothem characterization” to the end of that section. With this new arrangement, figures and tables have to the renumbered. 

R2: Guano Sampling: Table S2 gives us information on only some of the caves about whether they are active or inactive. However, nowhere can I find a complete list of the nine caves … therefore I cannot tell whether the guanos sampled were from active or inactive caves. Table 2 appears to lump them all together. Surely, a guano that is actively accumulating will have a somewhat different chemistry from a guano that is relict?

 7.>> We have now reorganized all supplementary tables, making the information on all samplings clearer. 

R2: I also have many issues with the discussion section … Line 328: “Acid solutions generated by the decomposition of guano and possible associated microbial activity produced various forms of corrosion in the floor and walls of those bat caves”. This raises several problems in my mind.

- Nothing whatsoever has been said in all of the results about forms of corrosion. These must be documented! … and we need to see that the corrosion forms in bat caves are different from those in non-bat caves.

8.>> As clarified above in answer 5, deep corrosion forms and speleothems like the ones present in bat caves were not observed in any non-bat cave. This is now mentioned along several parts in the text.

R2: - Since very little chemistry was done on waters from non-bat caves, how do we know that these do not also have acidic solutions? (an average pH of 4.1 for the non-bat waters might be considered to be rather acidic when compared to typical rain water pH of 5.7, and the pH values of the bat cave waters completely overlap the non-bat cave waters, and are not significantly different in a t-test.

9.>> Please check the new Table S2 in the Supplementary Material. We have seven samples from non-bat caves and 10 samples from superficial waters, including artificial lakes and drainages.

R2: Line 330: “bat caves were deeper, larger and bulkier”. Yes, this is shown by the numbers presented … but, as explained above, I feel that the choice of samples was not appropriate, and thus this point has not been adequately demonstrated.

10.>> Clarified above in answer 5, and a new Figure S1 is now provided.

R2: Line 331:... the bat caves “had more abundant, diversified and bigger speleothems” – is absolutely not supported by any evidence presented. That is largely because the section describing speleothems gave no indication of relative distribution, or relative diversity, or relative size for the two difference cave types.

11.>> We have now improved the description of speleothems in non-bat caves, stressing that corrosion forms like the ones present in bat caves were not observed in any non-bat cave (L214-218 and L311-314, plus legend in the new Figure 2).

R2: Line 331: “In an unprecedented example of bioengineering, we provided the first evidence that the long-term presence of bats (up to 23,000 years before present) and the guano deposits they produce mediated biological and chemical interactions which, by in turn, contributes to alter the geomorphology of those iron ore caves”. I have two comments:

- This is certainly not unprecedented! … There are now many publications reporting the erosional effects of bats/guano. This present study perhaps suggests a more significant role of bat guano in these particular caves, but similar studies have already been published. For example (one example of many), Figueira et al 2019 provide, for these same caves, analyses of speleothem, of guano, and of cave drip water, and attribute corrosion and deposition to bat guano … so this aspect of the present research is not unique. If there had been a clear demonstration of bat-v non-bat cave differences, then perhaps it might have been unique.

12.>> We used “unprecedented” in the sense of a bioengineering example, not about biocorrosion mediated by bat guano. As far as we know, this is the first study to clearly propose and detail the role of bats as bioengineers as the concept originally described by Jones et al, in 1994. In fact, in our Discussion we have acknowledged the role bats have in biocorrosion, citing several previous studies on this topic in the specific subsection “Bats and biogenic corrosion in caves - Operating mechanisms” (L467-509). The paper by Figueira et al. (2019) is indeed a very good suggestion of biocorrosion – and is now included in our references as #44 – but those authors did not address the bioengineering aspect like we did. Therefore, we sustain the use of “unprecedented” as we did. 

R2: - Long-term presence? Yes, they may have shown that some of the guano is quite old (but that is nothing new), but no evidence is presented that the caves with younger guano have different features from the caves with older guano. So, their dating is interesting enough, but does not prove that the duration of bat presence has any significance. 

13.>> Our radiocarbon dating data for the guano is the first for the entire Carajás region and, as far as we know, the second only published dating for guano in the entire Brazil. The information we present establishes temporal indicators of the use of caves by bats, confirming that this occupation dates back thousands of years, with one sample from the Pleistocene. We were prepared to use this information to infer further data on the speed of the corrosion process in these caves. Unfortunately, this was not possible: Luis Piló, the first author, was diagnosed with cancer in September 2021, preventing him from returning to the field for further analysis. Piló died 14 days after the manuscript was submitted for review. Therefore, the speed of the corrosion process in the caves we studied remains an open field for further analysis. 

R2: Here I have stopped reviewing the manuscript. The rest of the discussion is reasonable .. but it is based on inadequate data. While I think the hypothesis is good and most probably correct, this manuscript does not offer adequate proof. I am sorry that this manuscript is not more satisfactory. I think that the observations are interesting, but the research is poorly designed and cannot be published as it is.

14.>> We regret that the reviewer did not review the entire manuscript, treating our data as inadequate when, in fact, all clarifications to the points raised by him so far were duly clarified here. Perhaps it was not the Reviewer's intention, but we noticed a certain aggressiveness in some of the comments directed to our study.

R2: I lament that it has already been “published” in pre-print form.

15.>> With all due respect but with such a comment Reviewer 2 demonstrated that he/she is not familiar with preprints at all. We recommend him/her to check, for example, https://doi.org/10.1126/science.aaf9133 or https://doi.org/10.1371/journal.pcbi.1005473. There, Reviewer 2 will have the chance to better understand what is a preprint, their pros e cons. Moreover, he/she ignored that preprint is an option offered by PLOS to authors during the manuscript submission process. At the time of writing this answer, nearly 75 days after the publication of the preprint, its abstract has already been viewed 1,133 times, its pdf has been downloaded 168 times and the full article in html accessed 66 times. Such numbers demonstrate that our study has aroused the interest of thousands of people in a topic that often lacks media attention. So, we strongly believe that our decision to produce a preprint of our study cannot be considered regrettable and was part of a valid, correct and transparent scientific process.

Comments made by Reviewer 2 from the document “PONE-D-22-11101 Text with notes – Copy.pdf”

R2: - “If the impact is from presence of guano, and chemistry of guano, it is not really the bats that are the engineers (or any other organism producing guano). Does the effect work only with bat guano and not with, e.g., swiftlet guano, or oil bird guano? What about the feces of other organisms?”

16.>> Bat guano is a byproduct produced by bats (i.e., feces, resulted from ingested food items). Bats are, in this sense, the agents responsible for guano production, from ingesting insects, to processing them in their digestive tract, to transporting and defecating the guano into the caves. In the cave we studied, there are no other sources of guano in the volumes observed. No swiftlet, no oil birds. Moreover, the original definition of ecosystem engineers, by Jones et al. (1994) clearly states (our emphasis):

“Ecosystem engineers are organisms that directly or indirectly modulate the availability of resources to other species, by causing physical state changes in biotic or abiotic materials. In so doing they modify, maintain and create habitats. Autogenic engineers (e.g. corals, or trees) change the environment via their own physical structures (i.e. their living and dead tissues). Allogenic engineers (e.g. woodpeckers, beavers) change the environment by transforming living or non-living materials from one physical state to another, via mechanical or other means. The direct provision of resources to other species, in the form of living or dead tissues is not engineering. Organisms act as engineers when they modulate the supply of a resource or resources other than themselves.” Oikos 69, p. 373.

So, according to the original concept, bats are allogenic engineers, whose guano production is modifying and creating habitats in the caves we studied. Although sounds too obvious, it is not possible to have the observed bat guano without bats… and makes no sense to argue that the guano should be considered the engineers instead of the bats per se.

R2: - “This classification of allogenic does not seem to include chemical transformations”

17.>> In fact, it does. See the original definition provided above: “…via mechanical or other means.” 

R2: - “I contend that the bats themselves are not engineers if the effect is only from their guano. however, I do acknowledge that the originators of the concept included cows as engineers because their feces change the ecosystem of the field. So I suggest that it be clarified by inserting “(effected through their feces)”. It is not the bats themselves that are the key species since guano could be from other organisms and might have the same effects (e.g., oil birds, or swiftlets).”

18.>> Clarified above in answer 16. 

R2: - What criterion have you used to designate a cave as a bat cave or not? How many bats must a cave house to be classified as a bat cave?

19.>> Clarified above in answer 2.

Reviewer 1

One point, which is highlighted in the abstract and in the results, is missing according to me in the discussion is about the size of the caves. You wrote:” Caves currently harboring bat colonies and those with signs of past presence of such colonies had, on average, horizontal projections 4.5 times larger, areas 4.4 times larger, and volumes 5.0 times bigger than the reginal average ». This is a very interesting point but it is important to discuss it. Some people could ask if the bats didn’t choose the largest caves and you have to address this point. Of course, I understood that you are demonstrating how much the caves were enlarged thank to the biocorrosion and I agree. But to reinforce this point, it is needed to come back on these rates and discuss them here in the “Long-term effects of bat presence in caves » chapter. It can even give you some information, based on the erosion rate due to biocorrosion or for how long the caves were occupied by bats for example. Just an estimation which can bring you further than 22ky...

20.>> Addressed before in answers number 5 and 13.

---

## [Decision Letter · Decision Letter 1]

2 Sep 2022

PONE-D-22-11101R1

Bats as ecosystem engineers in iron ore caves in the Carajás National Forest, Brazilian Amazonia

PLOS ONE

Dear Dr. BERNARD,

Thank you for submitting your manuscript to PLOS ONE. After careful consideration, we have decided that your manuscript does not meet our criteria for publication and must therefore be rejected.

Specifically:

The decision is solely based on the policy of PLOS ONE regarding the data availability (speleothem characterization). As you probably know, detailed analyses and data availability are two of the seven publication criteria of this journal. Perhaps, you can either resubmit as a new publication after making the manuscript comply with these criteria or choose a journal more specific to zoology.

I am sorry that we cannot be more positive on this occasion, but hope that you appreciate the reasons for this decision.

Kind regards,

Il Won Kim

Academic Editor

PLOS ONE

Reviewers' comments:

Reviewer's Responses to Questions

**Comments to the Author**

1. If the authors have adequately addressed your comments raised in a previous round of review and you feel that this manuscript is now acceptable for publication, you may indicate that here to bypass the “Comments to the Author” section, enter your conflict of interest statement in the “Confidential to Editor” section, and submit your "Accept" recommendation.

Reviewer #1: All comments have been addressed

Reviewer #3: (No Response)

2. Is the manuscript technically sound, and do the data support the conclusions?

Reviewer #1: Yes

Reviewer #3: Partly

3. Has the statistical analysis been performed appropriately and rigorously? 

Reviewer #1: N/A

Reviewer #3: N/A

4. Have the authors made all data underlying the findings in their manuscript fully available?

Reviewer #1: Yes

Reviewer #3: No

5. Is the manuscript presented in an intelligible fashion and written in standard English?

Reviewer #1: Yes

Reviewer #3: Yes

6. Review Comments to the Author

Reviewer #1: Thank you to the authors to have taken in account my recomandation. They adressed my main comment according to the data they had. Things are clearer in this version. I recommand to accept this revised version as this is an original and precursor paper about biocorrosion in a very specific environment. No doubt that it will help a lot other researchers in this topic all around the world.

Reviewer #3: The authors appeared to respond to most points raised by the original two reviewers, especially on the overall description of the caves and their features. However, there are several experimental data missing as this reviewer examined the revised manuscript as a third reviewer.

In 'Materials and Methods', X-ray diffraction and X-ray spectrometry were mentioned. However, no real data were presented except the list of the minerals in 'Results: Speleothems characterization'. Without proper presentation of the data, it was impossible to review the characterization part of the manuscript. It must be challenging to systematically present the experimental evidence due to a large number of samples. Still, the lack of experimental data forced this reviewer to recommend the unfavorable decision, although the article is overall quite interesting.

Similarly, while microbial activity seems critical based on the authors' discussion, there is no experimental evidence to directly support the discussion.

There are several typos. To name a couple of them:

p11, final paragraph: M5SM2-0099 should read N5SM2-0099.

p17–: chemical formula for minerals. (e.g.) Fe3+(PO4) 2H2O looks unconventional. Usually, FePO4 2H2O.

7. PLOS authors have the option to publish the peer review history of their article (what does this mean?). If published, this will include your full peer review and any attached files.

Reviewer #1: **Yes: **BRUXELLES Laurent

Reviewer #3: No

- - - - -

---

## [Author Response · Author response to Decision Letter 1]

12 Sep 2022

Recife, Brazil, September 12th, 2022

Dear Miquel Vall-llosera Camps, Senior Editor 

After my appeal on the editorial decision for the manuscript PONE-D-22-11101R1, and following your instructions, I would like to submit a corrected version of the manuscript. Regarding Rviewer #3´s comments, these are our answers:

1. In 'Materials and Methods', X-ray diffraction and X-ray spectrometry were mentioned. However, no real data were presented except the list of the minerals in 'Results: Speleothems characterization'. Without proper presentation of the data, it was impossible to review the characterization part of the manuscript. It must be challenging to systematically present the experimental evidence due to a large number of samples.

>> In fact, as clearly stated in our Methodology, that part of the analysis was conducted at the University of São Paulo´s Escola Politécnica, in Brazil, a private lab hired for us to do that task. We have now attached as Supplementary Material all the official reports issued by all the laboratories we have hired (DRX, FRX, ICP, organic chemistry, and radio carbon analyses). 

2. Reviewer #3 indicated a "lack of experimental data" as the reason to recommend the unfavorable decision. 

>> Reviewer #3 did not present what lack of data is that. How can we, authors, contest this position if we don't even know what kind of data the reviewer is referring to? In case he/she is referring to the points raised by Reviewer #2, regarding expected matched samples (batcaves × non-bat caves), we once again emphasize the answers we provided before:

- We really would like to have had similar sample sizes (active bat caves x inactive bat caves x non-bat caves) to test our hypotheses in a statistically flawless approach, controlling all possible variables. However, in the real world this was not possible, since caves are unique and there are no two similar natural caves. 

- We demonstrated that bat caves are 0.2% of 1,309 caves analyzed in Carajás National Forest. Therefore, it would be impossible for us to have a larger number of bat caves to match the number of non-bat caves in the region.

- We demonstrated that cave size alone does not explain bat choice for these specific caves: Other caves as large as the largest bat cave studied [see Figure S1] have no signs of previous occupation by bats. 

- The deep corrosion signs, the richness and variety of speleothems described were observed only in bat caves (active or inactive). 

- Speleothems in caves with no expressive guano deposits were of local occurrence and small, while the greater diversity, abundance and size of speleothems was found only in the bat caves.

- Once again, this reinforces how unique and singular bat caves and their associated processes are, precluding us to have similar standardized and heterogeneous sample sizes. Insisting that we would have similar number of caves for the three categories we analyzed (active bat caves, inactive bat caves, non-bat caves) to address statistical issues is, therefore, impossible. 

3. Reviwer #3 indicated we had no experimental evidence to directly support the microbial activity. 

>> In fact, we have no experimental evidence because this was not our goal in this manuscript. In our discussion, we clearly stated that biological activity very likely has an important role in the processes we observed. Please check lines 426-457. In that part of our discussion, we present evidence produced by other authors, including evidence for other iron ore caves in Brazil. Testing biological activity is an open question in our field site and we had intention to approach that in the near future, but not in this paper. Therefore, rejecting our paper because of the lack of evidence based on an analysis that was not our objective – but was clearly and correctly presented, discussed and supported by references – seems unfair to us.

4. Reviewer #3 mentioned “several typos”, without clearly indicating them, except for: 

p11, final paragraph: M5SM2-0099 should read N5SM2-0099.

>> Corrected.

p17–: chemical formula for minerals. (e.g.) Fe3+(PO4) 2H2O looks unconventional. Usually, FePO4 2H2O.

>> We preserved the same right formulae provided in the reports issued by University of São Paulo´s Escola Politécnica. Please check the now included reports.

We have double-checked the entire manuscript for typos and did our best to detect and correct them.

Please note that our speleothem collections were all approved by the Centro Nacional de Pesquisa e Conservação de Cavernas CECAV, the federal agency responsible for that kind of permit. All our analysis were strictly under the law and we have now provided copies of the issued permits. 

Based on the answers provided above, please consider our revised version.

Thank you,

Enrico Bernard

---

## [Decision Letter · Decision Letter 2]

12 Jan 2023

PONE-D-22-11101R2

Bats as ecosystem engineers in iron ore caves in the Carajás National Forest, Brazilian AmazoniaP

PLOS ONE

Dear Dr. BERNARD,

Thank you for submitting your manuscript to PLOS ONE. After careful consideration, we feel that it has merit but does not fully meet PLOS ONE’s publication criteria as it currently stands. Therefore, we invite you to submit a revised version of the manuscript that addresses the points raised during the review process.

We look forward to receiving your revised manuscript.

Kind regards,

Ji-Zhong Wan

Academic Editor

PLOS ONE

Journal Requirements:

Additional Editor Comments:

Please solve all the issues purposed by the reviewers.

Request from the Editorial Staff:

We noted that a large quantity of rare speleothems, a non-renewable resource, were sampled for this study. Please provide ethical and scientific justification for the speleothem's sample size and a copy of your ethics documentation approving the collection of the speleothems (and an English translation, if the original is not in English language).

Reviewers' comments:

Reviewer's Responses to Questions

**Comments to the Author**

1. If the authors have adequately addressed your comments raised in a previous round of review and you feel that this manuscript is now acceptable for publication, you may indicate that here to bypass the “Comments to the Author” section, enter your conflict of interest statement in the “Confidential to Editor” section, and submit your "Accept" recommendation.

Reviewer #4: (No Response)

Reviewer #5: (No Response)

Reviewer #6: (No Response)

2. Is the manuscript technically sound, and do the data support the conclusions?

Reviewer #4: Partly

Reviewer #5: Yes

Reviewer #6: Yes

3. Has the statistical analysis been performed appropriately and rigorously? 

Reviewer #4: N/A

Reviewer #5: Yes

Reviewer #6: N/A

4. Have the authors made all data underlying the findings in their manuscript fully available?

Reviewer #4: Yes

Reviewer #5: Yes

Reviewer #6: Yes

5. Is the manuscript presented in an intelligible fashion and written in standard English?

Reviewer #4: Yes

Reviewer #5: Yes

Reviewer #6: Yes

6. Review Comments to the Author

Reviewer #4: This revised paper demonstrated the correlation between the presence of large bat colonies and the outstanding extension of caves developed in iron oxide rocks (BIF). Beyond this correlation, the most convincing arguments are the specific morphologies and speleothems developed in the bat caves. The experimental data which are now provided in suppl. Material clearly show the difference of mineralogical association, physical environment (pH…) in bat and non-bat caves.

The discussion mainly compares the processes in bat caves studied worldwide (by the way the literature review is quite comprehensive), with the morphologies and mineralogies identified in the studied area of Carajas. Based on their data and literature review, the authors statements (i.e. direct correlation between bats and intense geochemical processes of cave development) are convincing.

For this reason, such paper is worth publishing for its contribution to the knowledge of biocorrosion in a new environmental context (iron rocks). However, as stated by authors, the processes remain to be understood in detail, especially the microbial role, which should be a future direction.

Some small formal corrections must be addressed:

- The discussion between authors and previous reviewers about the way of writing chemical formula can be simply addressed by referring to the IMA conventional writings, which is the only acceptable way for minerals

- 202-203 : it is not clear which material has been dated on phosphate speleothems using 14C method. Phosphate or organic matter trapped in?

- Uncorrected spellings: 333 jarosite; 344 taranakite; 373 phosphosiderite

- 506 : why water percolating across iron is heated? Exothermic reaction??

- Fig 6 and 7 from the initial version have been removed in the revised version. However, from the caption, it seem these illustration could provide additional morphological consideration (fig 6), and a conceptual model (fig7), however I haven’t seen these figures…

Reviewer #5: In this research article the authors hypothesized that cave structures and corrosion processes observed in iron ore caves occupied for roosting purposes by large bat populations are unique and intrinsically mediated by the presence of bats and the guano they produce and, therefore, would not be observed in caves without bats. The study of the role of bats as ecosystem engineers was performed in the Carajás National Forest, Brazilian Amazonia, an area with > 1,500 caves, some holding ~150,000 bats. The authors analyzed the chemical composition and radiocarbon-dated guano deposits in bat caves to elucidate the time scale and chemical mechanisms involved in processes such as chemical deposition of speleothems and/or in the corrosion processes on the floors and walls of the caves. Comparison of active/inactive bat caves and non-bat caves suggests that acid solutions generated by the decomposition of guano and possible associated microbial activity produced various forms of corrosion, enlarging the cave, resulting in more abundant, diversified, and bigger speleothems. Though the study is descriptive, the manuscript is rather well written in a readable way. Argumentation and discussion of chemical mechanisms that may alter the geomorphology of iron ore caves is reasonable. It can be recommended for publication.

Minor comments

Keywords: “Biogenenic corrosion” should be revised as “Biogenic corrosion”

“bat nesting” should be revised as “bat roosting” on Line 526

“nesting bats” should be revised as “roosting bats” on Line 531

Reviewer #6: (No Response)

7. PLOS authors have the option to publish the peer review history of their article (what does this mean?). If published, this will include your full peer review and any attached files.

Reviewer #4: **Yes: **

Reviewer #5: No

Reviewer #6: No

---

## [Author Response · Author response to Decision Letter 2]

23 Jan 2023

Recife, Brazil, January 23rd 2023.

Dear Editor, 

Please find attached the corrected version of the manuscript PONE-D-22-11101R2 “Bats as ecosystem engineers in iron ore caves in the Carajás National Forest, Brazilian Amazonia”. We have addressed all the suggestions made by the reviewers. Bellow you will find a detailed lists of corrections. This was the fourth revision session of our manuscript and we hope that no other point is raised now so the manuscript can be finally published. Thank you very much.

Enrico Bernard 

Request from the Editorial Staff:

We noted that a large quantity of rare speleothems, a non-renewable resource, were sampled for this study. Please provide ethical and scientific justification for the speleothem's sample size and a copy of your ethics documentation approving the collection of the speleothems (and an English translation, if the original is not in English language).

>> As stated in our previous answers to the reviewers, we once again clarify that speleothems, including stalactites, stalagmites and crusts, were not collected in an unplanned, excessive or aleatory approach. Speleothems were collected from specific caves and, whenever possible, those speleothems collected provided multiple samples used for different purposes, including chemical analysis, radiocarbon dating and the preparation of cutting profiles for stratigraphic analyses. Several of the samples we analyzed provided first data for Brazil, justifying their collections. As specified before, Table S2 details the caves sampled and the methodology approach in each of them. A total of 18 stalactites, stalagmites and crusts were collected, producing 108 samples for analysis. Reviewer #1, which originally raised this concern, was likely induced to a misinterpretation because our original text was not clear enough. He very likely was induced to confound samples with collections. Anyways, we had clarified that in the next corrected version of the manuscript and, once again, here. All collections were strictly authorized by CECAV/ICMBio, the federal institution responsible for cave research and protection in Brazil. Besides the original legal permit, we have now attached an official letter from CECAV/ICMBio confirming that sample collections were in accordance with current legislation and authorized, with no objections, by the federal agency. A translated version of that letter is also presented.

Reviewers' comments:

Reviewer #4

1 - The discussion between authors and previous reviewers about the way of writing chemical formula can be simply addressed by referring to the IMA conventional writings, which is the only acceptable way for minerals.

>> Please note that, contrary to the previous reviewer which raised such point, our formulae were originally presented in strictly accordance with IMA conventional writings (http://cnmnc.units.it/). We have double checked and corrected three typos (incorrect space between elements, missing digit). All other formulae were originally correctly presented. 

2 - 202-203: it is not clear which material has been dated on phosphate speleothems using 14C method. Phosphate or organic matter trapped in?

>> The organic matter trapped in. Clarified now along the line.

3 - Uncorrected spellings: 333 jarosite; 344 taranakite; 373 phosphosiderite.

>> All corrected now. Thank you.

4 - 506 : why water percolating across iron is heated? Exothermic reaction??

>> No. The iron formation in Carajás is covered by iron-rich breccia generically known as canga, which act as caprock on plateau tops. Dripping is significant inside the caves due to their proximity to the surface and porosity of the canga and this layer tends to become quite hot due to the absorption of solar radiation. Therefore, the percolating water is heated not by exothermic reaction, but due to the higher temperature of the canga. 

5 - Fig 6 and 7 from the initial version have been removed in the revised version. However, from the caption, it seem these illustration could provide additional morphological consideration (fig 6), and a conceptual model (fig7), however I haven’t seen these figures…

>> Please note that Figs. 6 and 7 were not removed in the revised version. They are still present and this was probably a mistake caused by the uploading system.

Reviewer #5

1. Keywords: “Biogenenic corrosion” should be revised as “Biogenic corrosion”; “bat nesting” should be revised as “bat roosting” on Line 526; “nesting bats” should be revised as “roosting bats” on Line 531.

>> All corrected. Thank you.

---

## [Decision Letter · Decision Letter 3]

27 Apr 2023

Bats as ecosystem engineers in iron ore caves in the Carajás National Forest, Brazilian Amazonia

PONE-D-22-11101R3

Dear Dr. BERNARD,

We’re pleased to inform you that your manuscript has been judged scientifically suitable for publication and will be formally accepted for publication once it meets all outstanding technical requirements.

Kind regards,

Ji-Zhong Wan

Academic Editor

PLOS ONE

Additional Editor Comments (optional):

Reviewers' comments:

Reviewer's Responses to Questions

**Comments to the Author**

1. If the authors have adequately addressed your comments raised in a previous round of review and you feel that this manuscript is now acceptable for publication, you may indicate that here to bypass the “Comments to the Author” section, enter your conflict of interest statement in the “Confidential to Editor” section, and submit your "Accept" recommendation.

Reviewer #6: All comments have been addressed

2. Is the manuscript technically sound, and do the data support the conclusions?

Reviewer #6: Yes

3. Has the statistical analysis been performed appropriately and rigorously? 

Reviewer #6: N/A

4. Have the authors made all data underlying the findings in their manuscript fully available?

Reviewer #6: Yes

5. Is the manuscript presented in an intelligible fashion and written in standard English?

Reviewer #6: Yes

6. Review Comments to the Author

Reviewer #6: Dear Editor,

I have carefully reviewed this revised version of the manuscript and found that the authors have made all corrections and included the requested adjustments. Therefore, I recommend this manuscript for publication in its current format.

7. PLOS authors have the option to publish the peer review history of their article (what does this mean?). If published, this will include your full peer review and any attached files.

Reviewer #6: No

---

## [Editor Report · Acceptance letter]

2 May 2023

PONE-D-22-11101R3 

Bats as ecosystem engineers in iron ore caves in the Carajás National Forest, Brazilian Amazonia 

Dear Dr. Bernard:

I'm pleased to inform you that your manuscript has been deemed suitable for publication in PLOS ONE. Congratulations! Your manuscript is now with our production department. 

Kind regards, 

on behalf of

Dr. Ji-Zhong Wan 

Academic Editor

PLOS ONE